# Ribonucleotide reductase inhibitors suppress SAMHD1 ara-CTPase activity enhancing cytarabine efficacy

Sean G Rudd[1,*,‡] , Nikolaos Tsesmetzis[2,‡], Kumar Sanjiv[1,‡], Cynthia BJ Paulin[1,§], Lakshmi Sandhow[3], Juliane Kutzner[4], Ida Hed Myrberg[2] , Sarah S Bunten[4], Hanna Axelsson[5] , Si Min Zhang[1], Azita Rasti[1], Petri Mäkelä[1], Si'Ana A Coggins[6], Sijia Tao[6], Sharda Suman[1], Rui M Branca[7], Georgios Mermelekas[7], Elisée Wiita[1], Sun Lee[1], Julian Walfridsson[3], Raymond F Schinazi[6], Baek Kim[6,8], Janne Lehtiö[7] , Georgios Z Rassidakis[9], Katja Pokrovskaja Tamm[9], Ulrika Warpman-Berglund[1], Mats Heyman[2], Dan Grandér[9,†], Sören Lehmann[3,10], Thomas Lundbäck[5,11], Hong Qian[3], Jan-Inge Henter[2,12] , Torsten Schaller[4,¶] , Thomas Helleday[1,13] & Nikolas Herold[2,12,**]

## Abstract

The deoxycytidine analogue cytarabine (ara-C) remains the backbone treatment of acute myeloid leukaemia (AML) as well as other haematological and lymphoid malignancies, but must be combined with other chemotherapeutics to achieve cure. Yet, the underlying mechanism dictating synergistic efficacy of combination chemotherapy remains largely unknown. The dNTPase SAMHD1, which regulates dNTP homoeostasis antagonistically to ribonucleotide reductase (RNR), limits ara-C efficacy by hydrolysing the active triphosphate metabolite ara-CTP. Here, we report that clinically used inhibitors of RNR, such as gemcitabine and hydroxyurea, overcome the SAMHD1-mediated barrier to ara-C efficacy in primary blasts and mouse models of AML, displaying SAMHD1-dependent synergy with ara-C. We present evidence that this is mediated by dNTP pool imbalances leading to allosteric reduction of SAMHD1 ara-CTPase activity. Thus, SAMHD1 constitutes a novel biomarker for combination therapies of ara-C and RNR inhibitors with immediate consequences for clinical practice to improve treatment of AML.

**Keywords** acute myeloid leukaemia; chemotherapy resistance; drug synergy; precision medicine; SAMHD1
**Subject Categories** Cancer; Haematology

## Introduction

Five-year overall survival (OS) in AML varies with age, ranging from ~ 5% in elderly adults to more than 70% in children, causing more than 10,000 deaths yearly in the United States alone (De

1 Science for Life Laboratory, Department of Oncology-Pathology, Karolinska Institutet, Stockholm, Sweden
2 Childhood Cancer Research Unit, Department of Women's and Children's Health, Karolinska Institutet, Stockholm, Sweden
3 Center for Hematology and Regenerative Medicine, Department of Medicine, Karolinska University Hospital Huddinge, Karolinska Institutet, Stockholm, Sweden
4 Department of Infectious Diseases, Virology, University Hospital Heidelberg, Heidelberg, Germany
5 Chemical Biology Consortium Sweden, Science for Life Laboratory, Department of Medical Biochemistry and Biophysics, Karolinska Institutet, Stockholm, Sweden
6 Department of Pediatrics, Emory University School of Medicine, Atlanta, GA, USA
7 Department of Oncology-Pathology, Science for Life Laboratory, Karolinska Institutet, Stockholm, Sweden
8 Department of Pharmacy, Kyung-Hee University, Seoul, South Korea
9 Department of Oncology-Pathology, Karolinska Institutet, Stockholm, Sweden
10 Department of Medical Sciences, Uppsala University, Uppsala, Sweden
11 Mechanistic Biology & Profiling, Discovery Sciences, R&D, AstraZeneca, Gothenburg, Sweden
12 Paediatric Oncology, Theme of Children's Health, Karolinska University Hospital Solna, Stockholm, Sweden
13 Weston Park Cancer Centre, Department of Oncology and Metabolism, University of Sheffield, Sheffield, UK
*Corresponding author. Tel: +46 (0) 8 524 823 68; E-mail: sean.rudd@scilifelab.se
**Corresponding author. Tel: +46 (0) 8 524 832 04; E-mail: nikolas.herold@ki.se
†Deceased October 2017
‡These authors contributed equally to this work as first authors
§Present address: Research Institutes of Sweden (RISE), Material and Process Unit, Södertälje, Sweden
¶Present address: Heidelberg ImmunoTherapeutics GmbH, Heidelberg, Germany

Kouchkovsky & Abdul-Hay, 2016). Standard chemotherapy in AML treatment comprises anthracyclines, which are important for the achievement of complete remission during induction courses (Fernandez *et al*, 2009; Luskin *et al*, 2016), and the deoxycytidine analogue cytarabine (ara-C). The latter constitutes the backbone of high-dose remission consolidation therapy (Mayer *et al*, 1994; Lowenberg, 2013). The interpatient susceptibility to high-dose ara-C regimens is linked to the propensity of AML blasts to accumulate the active triphosphate metabolite ara-CTP (Plunkett *et al*, 1985), which causes DNA damage by perturbing DNA synthesis (Tsesmetzis *et al*, 2018). A main determinant for ara-CTP exposure, and thus a key factor for ara-C efficacy, is the deoxynucleoside triphosphate (dNTP) triphosphohydrolase SAM and HD domain-containing protein-1 (SAMHD1), which we and others identified as an ara-CTPase (Schneider *et al*, 2016; Herold *et al*, 2017a,b,c; Hollenbaugh *et al*, 2017; Rudd *et al*, 2017; Rassidakis *et al*, 2018). Accordingly, inactivation of SAMHD1 is a prime goal for rational improvement of ara-C-based therapies; however, no valid clinical strategies exist, nor are known efforts under development (Appendix Table S1).

# Results

### A phenotypic screen identifies gemcitabine as a SAMHD1-dependent ara-C sensitiser

As *in vitro* and *in silico*-based approaches thus far (Appendix Table S1) have not resulted in SAMHD1 inhibitors with sufficient cellular activity, we embarked upon a cell-based phenotypic screening strategy to identify such compounds (overview in Fig EV1A and B). We rationalised that a SAMHD1 inhibitor should sensitise SAMHD1-proficient cells to ara-C toxicity, but not their SAMHD1-deficient counterpart. We made use of our previously described pairs of CRISPR/Cas9-engineered THP-1 cells that differ with respect to their SAMHD1 status (Herold *et al*, 2017b) to measure the inhibitory effect on cell proliferation of sub-lethal ara-C concentrations in combination with a library of small molecules (for technical information, see Appendix Supplementary Methods and Appendix Fig S1). Briefly, in a first step, we screened a total of 33,467 compounds in SAMHD1-positive THP-1 cells in the presence of a sub-lethal concentration of ara-C (Fig EV1A and B, and Appendix Fig S1A and B). From the ~ 1,600 compounds that showed an inhibition of cell proliferation of ≥ 30% in the presence of ara-C, we then excluded compounds that were substantially toxic even in the absence of ara-C (Appendix Fig S1C), before performing concentration–response experiments for the remaining active substances in the presence and absence of ara-C in both SAMHD1-positive and SAMHD1-negative THP-1 cells (exemplified in Fig EV1C). We identified SAMHD1-independent ara-C sensitisers, such as the Wee1 inhibitor MK-1775, but also compounds demonstrating SAMHD1-dependent ara-C sensitisation. Amongst these, we were particularly intrigued by the clinically approved deoxycytidine analogue gemcitabine (dF-dC; Fig EV1C). Surprisingly, neither dF-dC itself nor its phosphorylated or deaminated metabolites inhibited SAMHD1 activity *in vitro* (Fig EV1D), and treatment of cells with dF-dC did not alter the thermal aggregation temperature ($T_{agg}$) of SAMHD1 (Fig EV1E), arguing against binding of dF-dC or an active metabolite thereof to SAMHD1 in living cells. These data indicate that the SAMHD1-dependent ara-C sensitisation by dF-dC was not due to a direct interaction with SAMHD1.

### Apparent suppression of SAMHD1 ara-CTPase by dF-dC is a result of RNR inhibition

The diphosphate metabolite of dF-dC (dF-dCDP) irreversibly inhibits the key enzyme in *de novo* dNTP synthesis, RNR (Cerqueira *et al*, 2007), which is consistent with an increased $T_{agg}$ of RRM1 observed in dF-dC-treated cells (Fig EV1E). Given the allosteric activation of SAMHD1 requires binding of (d)NTPs to two distinct sites on each monomer, interactions that are necessary for formation of the catalytically competent tetramer (Ji *et al*, 2013; Zhu *et al*, 2015), we hypothesised that alteration in the levels of these endogenous activators through RNR inhibition might be responsible for the apparent ablation of SAMHD1 ara-CTPase activity by dF-dC (Fig 1A). We thus predicted that other RNR inhibitors (RNRi) should also sensitise cells to ara-C in a manner dependent upon SAMHD1. To test this hypothesis, we treated a panel of SAMHD1-proficient or SAMHD1-deficient haematological cancer cell lines generated using CRISPR/Cas9 (Fig 1B) with a concentration–response matrix consisting of ara-C and an RNRi: either dF-dC, hydroxyurea (HU) or triapine (3-AP; Figs 1C and EV2A); none of which inhibited SAMHD1 *in vitro* (Fig EV1D). RNRi sensitised SAMHD1-proficient THP-1 cells to ara-C in a concentration-dependent manner, effectively reducing the half-maximal effective concentration ($EC_{50}$) for ara-C to that of their SAMHD1-deficient counterpart. However, ara-C sensitisation was consistently not observed in SAMHD1-deficient THP-1 cells (Fig 1C and D). Similar results were obtained with additional SAMHD1-proficient and SAMHD1-deficient cell lines of myeloid and lymphoid origin (Fig EV2A). Importantly, ectopic expression of wild-type (WT) SAMHD1, but not the catalytically inactive H233A mutant, could restore the RNRi-mediated ara-C sensitisation in SAMHD1-deficient THP-1 cells (Figs 1D and EV2A).

### SAMHD1 expression levels dictate the extent of synergy between ara-C and RNRi in cell lines

We subsequently performed drug synergy analyses using two reference models, highest single agent (HSA) (Berenbaum, 1989) (Fig EV2B) and zero interaction potency (ZIP) (Yadav *et al*, 2015) (Fig 1E). The HSA model defines synergy as a combinatorial effect that is larger than the individual drug effect observed at the same concentration, whilst the ZIP model combines the widely used Bliss independence and Loewe additivity models into a response surface model that uses a delta score to characterise synergy. According to both reference models, using the concentration–response matrix summary score, the interaction between ara-C and the RNRi was synergistic in all cell lines expressing dNTPase-proficient SAMHD1 (Figs 1E and EV2B). Either the absence of SAMHD1 protein or presence of catalytically inactive SAMHD1 abrogated synergy and significantly reduced the drug–drug interaction to a near-additive response (Figs 1E and EV2B). Notably, strongest synergy was observed in THP-1 cells, which express high levels of SAMHD1, whilst in HL-60 cells, with much lower SAMHD1 protein levels (Fig 1B), synergy was less pronounced (Figs 1E and EV2B). Consistently, analysis of a broader panel of

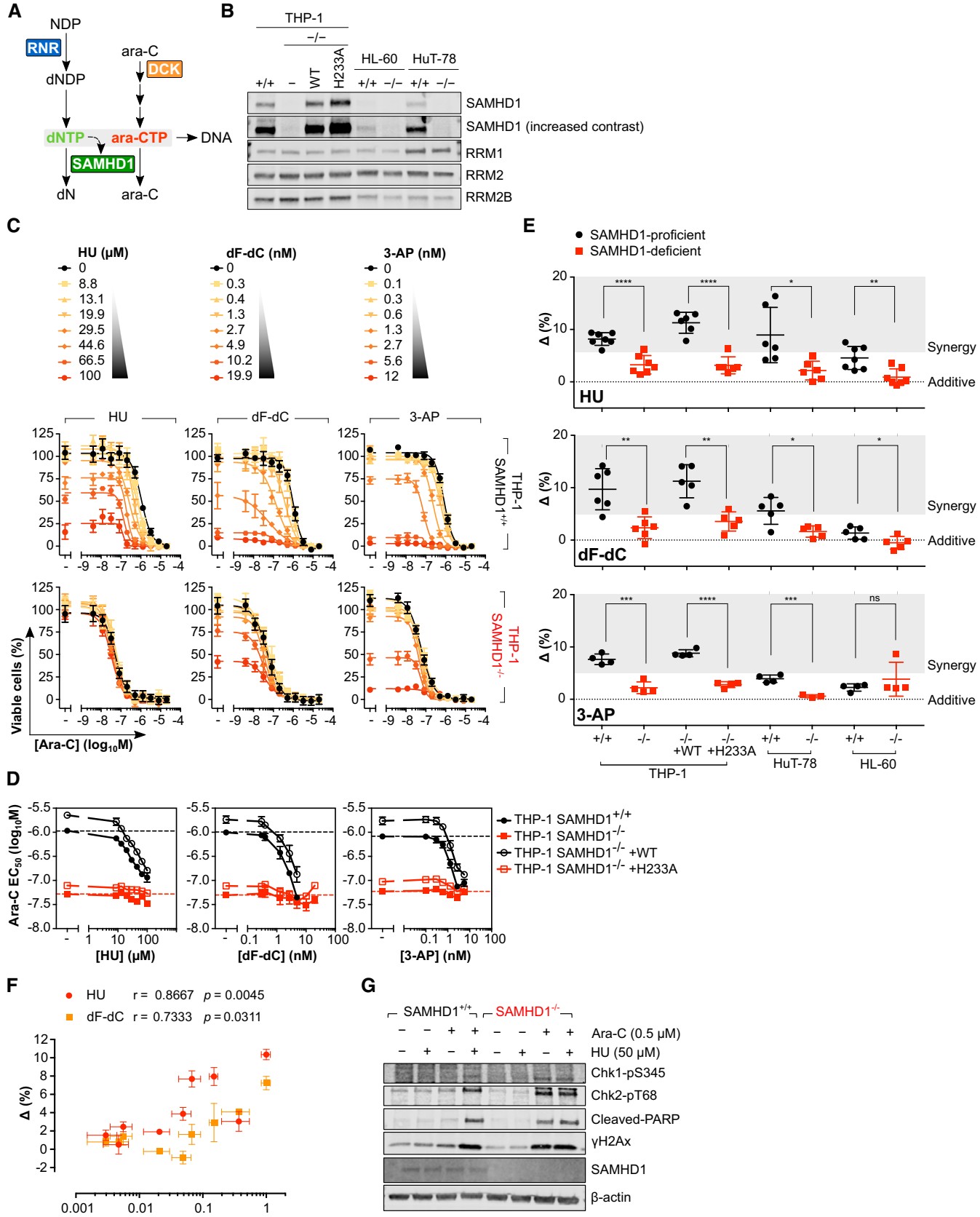

**Figure 1.**

**Figure 1. RNR inhibitor and ara-C synergy are dependent upon functional SAMHD1 in cancer cell models.**

A  Schematic detailing of proposed interplay between RNR and SAMHD1.
B  Immunoblot of lysates prepared from the indicated SAMHD1-proficient ($^{+/+}$), SAMHD1-deficient ($^{-/-}$) and rescue (WT, H233A) cell line pairs with the indicated antibodies. Representative of 2 independent experiments.
C  Proliferation inhibition analysis of ara-C and RNRi combination treatment in SAMHD1$^{+/+}$ or $^{-/-}$ THP-1 cells. Error bars indicate SEM of two (HU and dF-dC) or three (3-AP) independent experiments, each performed in duplicate.
D  Ara-C EC$_{50}$ values plotted as a function of RNRi concentration in SAMHD1$^{+/+}$, $^{-/-}$ and rescue (WT, H233A) THP-1 cell line pairs. EC$_{50}$ values in the absence of RNRi are indicated by the black and red dotted line. Error bars indicate SEM of two (HU and dF-dC) or three (3-AP) independent experiments, each performed in duplicate.
E  Drug synergy plots for ara-C and the indicated RNRi in SAMHD1$^{+/+}$, $^{-/-}$ and rescue (WT, H233A) cell line pairs. Each data point indicates an average delta score from a single dose–response matrix experiment performed in duplicate. Zero, > 0 or < 0 corresponds to additive, synergy or antagonism, respectively, whilst > 5 indicates strong synergy. The horizontal line and the error bars indicate the mean and SD, respectively, and statistical significance was determined using a two-tailed unpaired *t*-test: ns, not significant, $P \geq 0.05$; *$P < 0.05$; **$P < 0.01$; ***$P < 0.001$; ****$P < 0.0001$.
F  Spearman correlations of relative SAMHD1 protein abundance and synergy delta scores for ara-C versus HU or dF-dC in a panel ($n = 9$) of haematological cancer cell lines. Error bars indicate SEM. Each data point corresponds to SAMHD1 protein levels determined by immunoblot analysis ($n = 4$ for each cell line, representative blot shown in Appendix Fig S2) and an average delta score from repeated dose–response matrix experiments each performed in triplicate: THP-1, $n = 4$; HuT-78, $n = 2$; HL-60/iva, $n = 1$; KBM-7, $n = 2$ (HU) and 3 (dF-dC); K562, $n = 3$ (HU) and 4 (dF-dC); CCRF-CEM, $n = 3$ (HU) and 4 (dF-dC); MV-4-11, $n = 2$ (HU) and 3 (dF-dC); Jurkat, $n = 2$ (HU) and 3 (dF-dC); MOLT-4, $n = 2$ (HU) an 3 (dF-dC).
G  Immunoblot analysis of lysates prepared from SAMHD1$^{+/+}$ or $^{-/-}$ THP-1 cells treated for 24 h with ara-C and HU, as indicated. Representative of 3 independent experiments.

Source data are available online for this figure.

haematological cell lines revealed that synergy of dF-dC or HU with ara-C significantly correlated with SAMHD1 protein abundance (Fig 1F, Appendix Fig S2). In accordance, sub-lethal concentrations (EC$_{10}$) of ara-C alone caused no induction of DNA damage signalling in SAMHD1-proficient cells, whilst the combination with sub-toxic doses of HU (Fig 1G) or dF-dC (Fig EV2C) led to robust DNA damage (indicated by Chk1-pS345, Chk2-pT68 and γH2Ax) and apoptotic (indicated by cleaved polyADP-ribose polymerase [PARP]) signalling, to an extent similar to low-dose ara-C alone in SAMHD1-deficient cells.

## Allosteric inhibitors of RNR do not synergise with ara-C in a SAMHD1-dependent manner

Purine nucleoside analogues are clinically combined with ara-C. Some have also been shown to be substrates/activators of SAMHD1 (Arnold *et al*, 2015b; Herold *et al*, 2017a; Hollenbaugh *et al*, 2017; Knecht *et al*, 2018) and, importantly, are documented to allosterically inhibit RNR as part of their cytotoxic mechanism (Aye & Stubbe, 2011; Wisitpitthaya *et al*, 2016). We thus tested clofarabine (Cl-F-ara-A), fludarabine (2-F-ara-A) and cladribine (2-CdA) for their ability to synergise with ara-C. Unlike the previously tested non-allosteric RNRi, allosteric RNRi Cl-F-ara-A and 2-F-ara-A displayed only weak synergy with ara-C, whilst 2-CdA synergised strongly, consistent with previous reports (Chow *et al*, 2003; Stumpel *et al*, 2015). However, no SAMHD1-dependent drug–drug interaction was observed across the cell line panel (Fig EV3A–D; and possible reasons for this are detailed in the Discussion). The Wee1 kinase inhibitor MK-1775, which we identified as a SAMHD1-independent ara-C sensitiser (Fig EV1C), has been described as an enhancer of ara-C toxicity before (Van Linden *et al*, 2013). We confirmed this in our drug combination matrix experiments, showing that this synergy was largely independent of the SAMHD1 status, even though the degree of synergy was more pronounced in some SAMHD1-positive cell models (Appendix Fig S3). We also tested histone deacetylase inhibitors (HDACi) in our drug combination matrix experiments given HU can mimic some cellular effects of HDACi at least in sickle cell disease (Cao, 2004). However, in

contrast to HU and other non-allosteric RNRi, only weak ara-C sensitisation was observed, and this was independent of SAMHD1 status (Appendix Fig S4).

Given ara-C is routinely combined with an anthracycline during AML treatment, we next performed drug combination matrix experiments in which either daunorubicin or doxorubicin was added to combinations of HU and ara-C. Whilst toxic concentrations of either anthracycline decreased cell viability, the ability of HU to sensitise cells to ara-C in SAMHD1-positive, but not SAMHD1-negative, cells was preserved (Appendix Fig S5).

Taken together, these data indicate that a SAMHD1-dependent synergism with ara-C is most pronounced with non-allosteric RNRi such as dF-dC, HU and 3-AP, but not with allosteric purine nucleoside RNRi (e.g. Cl-F-ara-A, 2-F-ara-A and 2-CdA), the Wee1 kinase inhibitor MK-1775, or HDACi. Of interest to potential clinical application, the SAMHD1-dependent synergy of non-allosteric RNRi and ara-C is not affected by the concomitant treatment with anthracyclines.

## RNR inhibition relieves the SAMHD1-mediated barrier to ara-C treatment *in vivo*

Next, we sought to investigate whether inhibition of RNR would alleviate the SAMHD1-mediated resistance to ara-C *in vivo*. As HU has been used in the treatment of AML for decades, is devoid of dF-dC induced toxicity associated with repetitive dosing (O'Rourke *et al*, 1994) and is furthermore cheap and highly accessible, we decided to focus on this RNRi in two orthotopic mouse models of AML. First, we injected either SAMHD1$^{-/-}$ or SAMHD1$^{+/+}$ THP-1 cell clones carrying a luciferase reporter into the tail vein of NOD/SCID mice subsequent to treatment with PBS or ara-C and HU, alone or in combination (Fig 2A, Appendix Fig S6). Irrespective of SAMHD1 status, mice treated with PBS only developed signs of systemic disease after ~ 35 days and succumbed after a median time span of 50 days. In SAMHD1-proficient AML, ara-C treatment had no effect on survival as compared to PBS treatment, but ara-C significantly prolonged survival in SAMHD1-deficient AML mice, resulting in a median survival of 68 days

($P$ = 0.0018), consistent with our previously published results (Herold *et al*, 2017a,b,c). Combination treatment of HU and ara-C in mice xenotransplanted with SAMHD1-proficient THP-1 cells resulted in a median survival of 64 days, significantly better than ara-C treatment alone ($P$ = 0.0141). These results were recapitulated in a second experiment using SAMHD1-proficient and SAMHD1-deficient HL60/iva clones (Fig 2B, Appendix Fig S7). In both experiments, combination treatment caused transient weight loss in mice (Appendix Figs S6D and E, and S7D and E). Of note, in both studies, a trend towards improved survival was observed in mice with SAMHD1-deficient AML cells when comparing combination treatment with ara-C treatment alone ($P$ = 0.0737 and $P$ = 0.0893).

To complement this dataset, we performed an additional experiment using the RNRi dF-dC in the THP-1 SAMHD1$^{+/+}$ AML mouse model. To mitigate dF-dC toxicity associated with repetitive dosing as described above, we only administered two doses of dF-dC on days 1 and 3 of the 5-day treatment regimen. Median survival in this experiment did not significantly differ for animals treated with PBS, ara-C or dF-dC (44, 47 and 49.5 days, respectively, Fig 2C, Appendix Fig S8). However, combination of ara-C with dF-dC led to a median survival of 65 days, significantly longer as compared to ara-C or dF-dC alone ($P$ = 0.0014, and $P$ = 0.0097, respectively, Fig 2C). Also in this experiment, transient weight loss was observed in the combination treatment (Appendix Fig S8B). Taken together, these data demonstrate that combining the RNRi HU or dF-dC with ara-C can overcome the SAMHD1-mediated barrier to ara-C efficacy *in vivo*.

As xenotransplantation of human cells requires the use of immunocompromised mice, we next employed a syngeneic murine AML model using myeloid precursors transformed by transduction with the fusion gene *MLL-AF9* (Xiao *et al*, 2018) to further assess the combination of ara-C and HU. *MLL-AF9*-transformed blasts showed detectable expression of SAMHD1 and, in accordance, could be moderately sensitised to ara-C by HU *in vitro* (Appendix Fig S9). Median survival for this aggressive AML model treated with normal saline (NS, vehicle), HU, ara-C or the combination of ara-C and HU was 6, 8, 12 and 14 days post-treatment, respectively (Fig 2D). Significance in the difference of survival was reached comparing ara-C and HU with vehicle ($P$ = 0.0026), but not comparing ara-C only or HU only with vehicle ($P$ = 0.0995, and $P$ = 0.2252, respectively). This model allowed the parallel study of myelotoxicity, which is relevant as both ara-C and HU are myelotoxic drugs, and excessive bone marrow toxicity might complicate the use of this combination treatment in clinical settings. On day 1 post-chemotherapy, whilst both ara-C and ara-C plus HU significantly reduced the total white blood count (WBC), red blood cell count (RBC), haemoglobin and mean corpuscular volume (MCV) of erythrocytes in peripheral blood as compared to vehicle, no significant differences in these parameters were observed comparing ara-C with ara-C and HU (Fig EV4A–D). In addition, mean corpuscular haemoglobin (MCH) and platelet counts were not affected adversely (Fig EV4E and F). Similarly, no significant differences in bone marrow cellularity and spleen weight at sacrifice were measured (Fig EV4G and H). This suggests that myelotoxicity of ara-C and HU combination therapy is not in excess of myelotoxicity of ara-C alone in immunocompetent mice.

## SAMHD1 expression levels dictate the extent of synergy between ara-C and RNRi in primary patient-derived AML blasts

To determine whether the RNRi HU and dF-dC synergise with ara-C in primary patient cells, we subjected adult ($n$ = 8) and paediatric ($n$ = 8) AML blasts *ex vivo* to concentration–response matrices of ara-C and HU or dF-dC. In the majority of patient samples, with increasing doses of either dF-dC or HU, increased sensitivity to ara-C was observed (Appendix Fig S10A and B). Accordingly, determination of summary synergy scores using both ZIP and HSA reference models indicated synergy of RNRi and ara-C in the majority of samples (Fig 3A, Appendix Fig S10D). After performing quantitative immunoblotting of SAMHD1 from lysates prepared from the same patient blasts (Appendix Fig S10C), we revealed that the extent of synergy using the ZIP reference model significantly correlated with the abundance of SAMHD1 protein ($r$ = 0.4189; $P$ = 0.0466; Fig 3B). A similar trend was observed using the HSA model (Appendix Fig S10E). To further interrogate the dependence of RNRi and ara-C synergy upon catalytically active SAMHD1, we pre-treated patient AML blasts *ex vivo* with virus-like particles (VLPs) either containing (X) or lacking (dX) the lentiviral protein Vpx that depletes SAMHD1 by targeting it for proteasomal degradation, prior to incubating them with ara-C and RNRi concentration–response matrices. As demonstrated previously (Hrecka *et al*, 2011; Laguette *et al*, 2011; Herold *et al*, 2017b), Vpx treatment efficiently depleted SAMHD1 protein (Fig 3C and E, Appendix Fig S10C) and, in line with our data in cancer cell lines, completely abolished the RNRi-mediated sensitisation to ara-C toxicity (Fig 3D and F). With increasing doses of HU or dF-dC, concentration-dependent decreases in ara-C EC$_{50}$ values were observed in the majority of samples evaluated, in some cases reducing the ara-C EC$_{50}$ values by two orders of magnitude (Fig 3F). Comparison of the summary synergy scores of those samples treated with Vpx-VLPs or control VLPs, using the ZIP reference model, revealed a significant ($P$ = 0.0046) reduction in the extent of synergy, from a median delta score of 8.5 to 3.05 (Fig 3G), and a similar result was obtained using the HSA reference model (Appendix Fig S10F). These data are in agreement with the results obtained in cancer cell lines (Fig 1F), demonstrating that the level of SAMHD1 dictates the extent of RNRi and ara-C synergy.

## High expression of SAMHD1 and RNR subunits correlates with reduced survival in AML patients

Given that inhibitors of RNR activity modulated ara-C toxicity in a SAMHD1-dependent manner, we next evaluated whether expression levels of RNR genes impact the survival of AML patients treated with ara-C. We re-assessed clinical data and mRNA expression levels of patients treated with ara-C from the publicly available adult *de novo* and paediatric AML databases from the The Cancer Genome Atlas (TCGA) and the Therapeutically Applicable Research To Generate Effective Treatments (TARGET) projects, respectively, as described previously (Herold *et al*, 2017b). Only *RRM2B* encoding the p53-induced small subunit of RNR showed statistically significantly higher hazard ratios (HRs) for event-free (EFS) and OS for ara-C-treated AML patients in univariable Cox proportional hazard regression analyses. However, when analysed

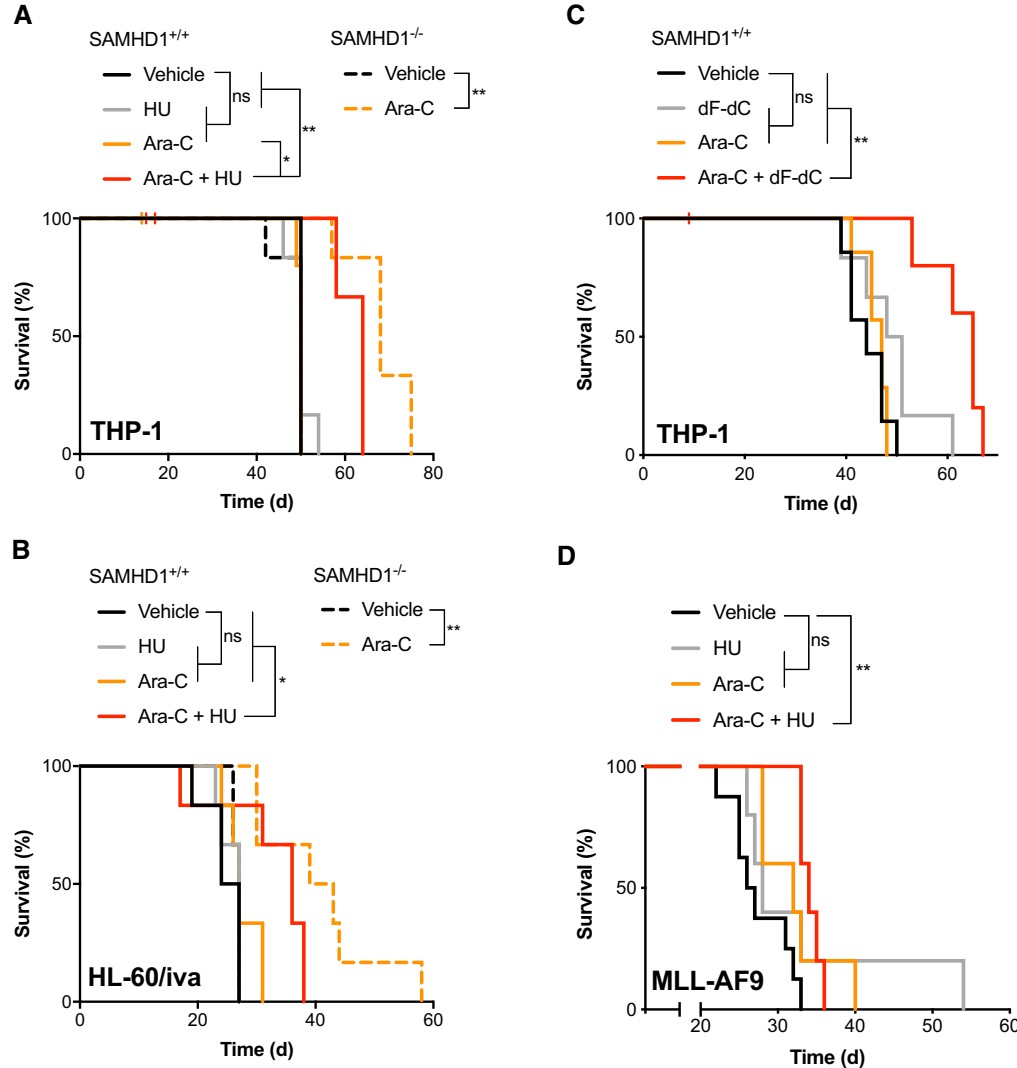

**Figure 2. RNR inhibition overcomes the SAMHD1-mediated barrier to ara-C in AML mouse models.**

A  Kaplan–Meier analysis of NOD/SCID mice injected i.v. with luciferase-expressing SAMHD1$^{+/+}$ or $^{-/-}$ THP-1 cell clones (day 0) and treated with ara-C and/or HU as indicated (day 6). $n$ = 6 per treatment group. For further data and analysis, see Appendix Fig S6.

B  Kaplan–Meier analysis of NOD/SCID mice injected i.v. with luciferase-expressing SAMHD1$^{+/+}$ or $^{-/-}$ HL-60/iva cell clones (day 0) and treated with ara-C and/or HU as indicated (day 6). $n$ = 6 per treatment group. For further data and analysis, see Appendix Fig S7.

C  Kaplan–Meier analysis of NOD/SCID mice injected i.v. with luciferase-expressing SAMHD1$^{+/+}$ THP-1 cell clone (day 0) and treated with ara-C and/or dF-dC as indicated (day 6). $n$ = 7 per treatment group. For further data and analysis, see Appendix Fig S8.

D  Kaplan–Meier analysis of CD45.2 C57BL/6J mice injected i.v. with murine *MLL-AF9*-transformed AML blasts (day 0) and treated with ara-C and/or HU days 20–24. $n$ = 5 per treatment group, except for vehicle ($n$ = 4). For further data and analysis, see Appendix Figs S8 and S9.

Data information: Tick marks indicate censored animals. Statistical significance determined using Mantel–Cox log-rank test: ns, not significant, $P \geq 0.05$; *$P < 0.05$; **$P < 0.01$.

Source data are available online for this figure.

---

in the same model as *RRM1*, *RRM2* and *RRM2B*, respectively (Table 1), *SAMHD1* showed slight increases of HRs in multivariable regression in particular for OS after 18 and 12 months for the TCGA and TARGET cohorts, respectively. Importantly, significance was maintained despite a loss of power as compared to univariable analyses. This is consistent with the notion that the interplay between SAMHD1 and RNR is important for the efficacy of ara-C therapies.

## RNRi invert the ratio of dCTP-to-dATP concentrations and activate dCK

Thus far, we have established that RNRi can sensitise cells to ara-C in a SAMHD1-dependent manner, albeit without directly inhibiting SAMHD1. The activity of SAMHD1 can be regulated by post-translational modifications, and so we speculated this could be the cause of the apparent loss of ara-CTPase activity. Reactive oxygen

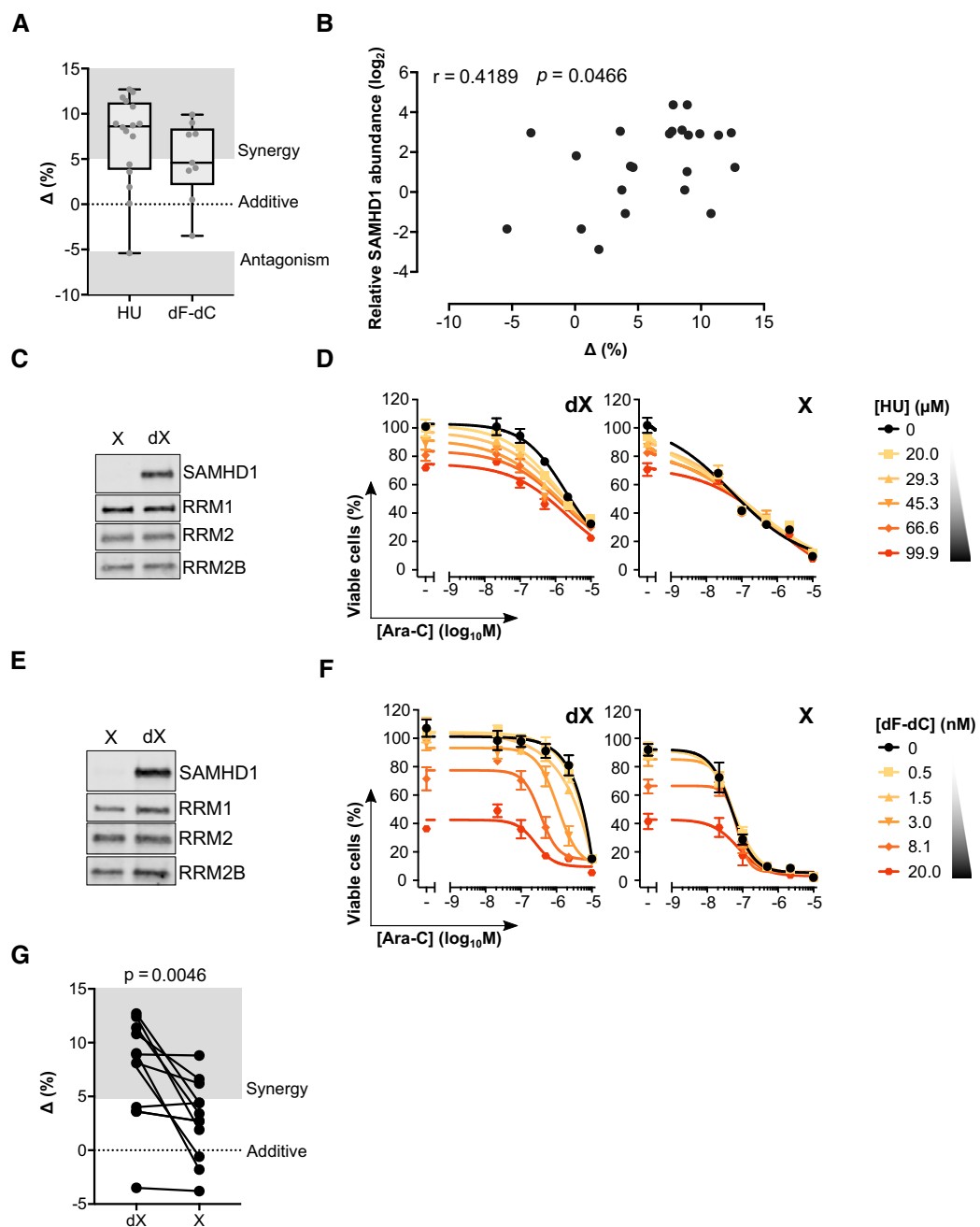

**Figure 3. RNR inhibition enhances ara-C efficacy in primary patient AML blasts in a SAMHD1-dependent manner.**

A   Drug synergy plots for ara-C and HU or dF-dC in primary patient-derived AML blasts. Each data point indicates an average delta score from a single patient sample subjected to a dose–response matrix experiment performed in triplicate, $n = 16$ for HU and $n = 9$ for dF-dC. Zero, > 0 or < 0 corresponds to additive effects, synergy or antagonism, respectively, whilst > 5 indicates strong synergy. Median, upper and lower quartiles, and range of delta scores are indicated by box-and-whisker plots. For proliferation inhibition curves for each sample, see Appendix Fig S10A and B, and for patient characteristics, see Appendix Table S2.

B   Pearson correlation of relative SAMHD1 protein abundance and synergy delta scores for ara-C and HU or dF-dC in primary patient-derived AML blasts ($n = 23$). For immunoblot analysis of SAMHD1 protein abundance, see Appendix Fig S10C.

C–F  Immunoblot of primary patient-derived AML blasts treated with control (dX) or Vpx-containing (X) virus-like particles (VLPs): patient A2953 (C), ALG17_001 (E). Accompanying proliferation inhibition analysis of ara-C and indicated RNRi combination in these samples: patient A2953 (D), ALG17_001 (D). Error bars indicate SD of single experiment performed in triplicate.

G   Paired drug synergy plot for ara-C and RNRi (HU, $n = 7$; dF-dC, $n = 5$) in primary patient-derived AML blasts pre-treated with control (dX) or Vpx-containing (X) VLPs. Zero, > 0 or < 0 corresponds to additive effects, synergy or antagonism, respectively, whilst > 5 indicates strong synergy and < 5 indicates strong antagonism. Each data point indicates an average delta score from a single patient sample subjected to a dose–response matrix experiment performed in triplicate. Statistical testing was performed using two-way ANOVA.

Source data are available online for this figure.

Table 1.  Hazard ratios (HR) for mRA levels of *SAMHD1* and *RRM1*, *RRM2* and *RRM2B* (all log-transformed using the natural logarithm) in univariable regression as well as hazard ratios for SAMHD1 in multivariable regression models in ara-C-treated AML patients.

| TCGA cohort | | | | |
|---|---|---|---|---|
| mRNA | EFS[a,h,i] complete[c] | EFS 18 months[d] | OS[b,h,i] complete | OS 18 months |
| Univariable | | | | |
| *SAMHD1* | **1.16 (1.00–1.34; 0.0419)** | **1.23 (1.05–1.46; 0.0101)** | 1.15 (0.99–1.34; 0.0628) | **1.25 (1.03–1.53; 0.0257)** |
| *RRM1*[e] | 1.55 (0.99–2.43; 0.0553) | 1.42 (0.87–2.33; 0.1652) | 1.58 (0.99–2.50; 0.0504) | 1.49 (0.83–2.66; 0.1833) |
| *RRM2*[f] | 1.23 (0.89–1.73; 0.2065) | 1.25 (0.88–1.80; 0.2142) | 1.11 (0.78–1.57; 0.5727) | 1.15 (0.75–1.76; 0.5206) |
| *RRM2B*[g] | 1.69 (0.95–3.00; 0.0753) | 1.59 (0.85–2.99; 0.1459) | **1.86 (1.03–3.36; 0.0361)** | 1.33 (0.64–2.81 (0.4424) |
| Multivariable (*SAMHD1*) | | | | |
| *SAMHD1* + *RRM1* | **1.18 (1.02–1.36; 0.0293)** | **1.25 (1.06–1.48; 0.0080)** | **1.17 (1.00–1.37; 0.0442)** | **1.27 (1.04–1.56; 0.0209)** |
| *SAMHD1* + *RRM2* | 1.15 (0.99–1.33; 0.0612) | **1.23 (1.04–1.45; 0.0158)** | 1.15 (0.98–1.34; 0.0714) | **1.25 (1.02–1.53; 0.0299)** |
| *SAMHD1* + *RRM2B* | **1.15 (1.00–1.33; 0.0455)** | **1.23 (1.04–1.45; 0.0116)** | 1.15 (0.99–1.33; 0.0603) | **1.25 (1.02–1.52; 0.0286)** |
| *SAMHD1* + *RRM1* + *RRM2B* | **1.17 (1.01–1.35; 0.0329)** | **1.24 (1.05–1.46; 0.0094)** | **1.16 (1.00–1.36; 0.0458)** | **1.27 (1.03–1.55; 0.0225)** |
| TARGET cohort | | | | |
| mRNA | EFS complete | EFS 12 months[d] | OS complete | OS 12 months |
| Univariable | | | | |
| *SAMHD1* | 1.00 (0.87–1.17; 0.9547) | 1.01 (0.83–1.23; 0.9321) | 0.96 (0.80–1.16; 0.6870) | **1.54 (1.02–2.31; 0.0381)** |
| *RRM1* | 1.38 (0.91–2.08; 0.1301) | 1.49 (0.84–2.63; 0.1724) | 1.49 (0.89–2.51; 0.1332) | 0.91 (0.31–2,67; 0.8567) |
| *RRM2* | 1.09 (0.90–1.32; 0.3671) | 1.19 (0.90–1.58; 0.2224) | 1.09 (0.85–1.40; 0.4886) | 1.41 (0.80–2.48; 0.2368) |
| *RRM2B* | **1.39 (1.04–1.85; 0.0279)** | 1.28 (0.86–1.91; 0.2176) | **1.47 (1.01–2.14; 0.0416)** | 0.79 (0.37–1.68; 0.5406) |
| Multivariable (*SAMHD1*) | | | | |
| *SAMHD1* + *RRM1* | 1.01 (0.87–1.17; 0.9294) | 1.02 (0.83–1.24; 0.8700) | 1.01 (0.87–1.17; 0.9294) | **1.54 (1.02–2.31; 0.0384)** |
| *SAMHD1* + *RRM2* | 0.97 (0.82–1.14; 0.7188) | 0.95 (0.77–1.18; 0.6555) | 0.97 (0.82–1.14; 0.7189) | 1.48 (0.95–2.29; 0.0803) |
| *SAMHD1* + *RRM2B* | 1.09 (0.92–1.28; 0.3157) | 1.07 (0.86–1.34; 0.5267) | 1.09 (0.92–1.28; 0.3157) | **1.56 (1.01–2.41; 0.0461)** |
| *SAMHD1* + *RRM1* + *RRM2B* | 1.08 (0.91–1.27; 0.3587) | 1.06 (0.86–1.32; 0.5721) | 1.08 (0.92–1.27; 0.3587) | **1.56 (1.01–2.43; 0.0466)** |

[a]Event-free survival.
[b]Overall survival.
[c]Complete follow-up period.
[d]Follow-up censored after the first 18 or 12 months after diagnosis.
[e]Ribonucleoside-diphosphate reductase (large) subunit M1.
[f]Ribonucleoside-diphosphate reductase (small) subunit M2.
[g]Ribonucleoside-diphosphate reductase (small) subunit M2B.
[h]Adjusted for age, sex and cytogenetic risk group.
[i]Shown are hazard ratios, 95% confidence intervals and *P*-values calculated with Wald test. Bold text indicates *P*-values < 0.05.

species (ROS) can reversibly oxidise cysteine residues in SAMHD1 resulting in inhibition of tetramerisation and catalysis (Mauney *et al*, 2017), and RNR inhibition is known to induce ROS (Somyajit *et al*, 2017; Patra *et al*, 2019). However, pre-treatment of cells with the ROS scavenger *N*-acetylcysteine (NAC) had no effect on synergy between ara-C and HU (Appendix Fig S11). Threonine phosphorylation at position 592 (T592) of SAMHD1 by cyclin-dependent kinases 1 and 2 (Cdk1/2) has also been implicated in regulating catalytic activity, in particular when dNTP levels are low (Arnold *et al*, 2015a; Yan *et al*, 2015). RNR inhibition can perturb cell cycle progression and thereby affect expression and/or activity of Cdk1/2 (Rieber & Rieber, 1994; Tanguay & Chiles, 1994; Rodriguez-Bravo *et al*, 2007). However, expression of a phosphomimetic T592E or phosphorylation-null T592A mutant SAMHD1 in SAMHD1-deficient THP-1 cells had little effect upon

the ability of HU or dF-dC to sensitise these cells to ara-C (Appendix Fig S11).

Next, we hypothesised that the apparent loss of SAMHD1 ara-CTPase activity may be due to perturbation of dNTP pools, which are required for allosteric activation of the ara-CTPase activity of SAMHD1 (Fig 1A). Depletion of dNTP pools might push the effective concentrations of allosteric activators at the second allosteric site (AS2) below a threshold required to maintain the catalytically competent tetrameric conformation. Hence, we investigated whether the relative composition of monomers, dimers and tetramers of SAMHD1 was affected by inhibition of RNR using *in vivo* cross-linking experiments. Surprisingly, no gross changes in the proportion of tetrameric SAMHD1 were observed in HU- or dF-dC-treated cells (Fig EV5A–D). These data are supported also by the lack of a substantial change in the $T_{agg}$ of SAMHD1 in HU- or dF-dC-treated

cells (Fig EV5E–F), which would otherwise indicate a change in oligomeric composition, as demonstrated by an oligomerisation-dead mutant of SAMHD1 with a greatly reduced $T_{agg}$ (Fig EV5G). Therefore, we concluded that treatment of cells with an RNRi did not greatly alter the oligomeric structure of SAMHD1. This argues against a depletion of allosteric AS2 activators as the underlying cause of the indirect loss of ara-CTPase activity observed following RNR inhibition.

To assess the effects of RNRi on dNTP pools directly, THP-1 cells were treated with low doses of either HU or 3-AP, and individual dNTP species measured using a primer extension assay (Diamond *et al*, 2004). Differential effects on purine and pyrimidine dNTP pools were observed (Appendix Fig S12A–D); irrespective of treatment, dTTP was clearly the most abundant dNTP species, whilst the least abundant under untreated conditions, dCTP, reached similar levels as dGTP and surpassed those of dATP following RNRi treatment. As a net result, dCTP-to-dATP ratios were not only increased threefold to sixfold, but also inverted from $0.6 \pm 0.1$ to up to $3.5 \pm 0.8$ by RNRi treatment (Fig 4A, Appendix Fig S12A–D). Next, we performed combination experiments with ara-C and either HU, 3-AP or dF-dC using liquid chromatography–mass spectrometry (LC-MS/MS) (Fromentin *et al*, 2010)—as this allowed the incorporation of ara-C and dF-dC into the experiments (whose triphosphate metabolites would interfere with the primer extension assay). Intracellular amounts of ara-CTP when adding an RNRi to ara-C in SAMHD1-proficient THP-1 cells were significantly increased by a factor of $\sim 4$ as compared to treatment with ara-C alone, almost achieving levels of ara-CTP in SAMHD1-deficient THP-1 cells treated with ara-C (Fig 4B). This correlated with an increased dCTP-to-dATP ratio in RNRi-treated cells that was unaffected by treatment with ara-C alone (Fig 4B).

As RNRi HU, dF-dC and 3-AP inhibited *de novo* synthesis of dNTPs, we hypothesised that the differential net effects on dNTP species might stem from a concomitant activation of the dNTP salvage pathway. Consistent with this, RNRi treatment led to an increase in activating phosphorylation of salvage enzyme dCK at serine-74 $\sim 8$- to 20-fold (Fig 4C).

### dCTPαS-activated SAMHD1 is a poor ara-CTPase

Thus far, we have established that RNRi do not cause a net reduction of SAMHD1 tetramers and lead to dNTP imbalances rather than absolute depletion that nonetheless correlated with a concomitant increase in ara-CTP. Next, we investigated whether the ara-CTPase activity of SAMHD1 is differentially activated by the dNTP occupying AS2, which could explain the presence of a tetrameric yet ara-CTPase-deficient SAMHD1. Conducting biochemical experiments, we incubated recombinant SAMHD1 with saturating concentrations of GTP (as an allosteric regulator for AS1), ara-CTP (as a substrate for the catalytic site) and a titration of a series of dNTPαS, i.e. non-hydrolysable dNTP analogues intended as allosteric regulators for the AS2 site. Whilst dGTPαS, dATPαS and dTTPαS could activate the hydrolysis of ara-CTP, dCTPαS, even at concentrations up to 20- to 100-fold higher than needed for saturation of ara-CTPase activity with the other dNTP analogues, could not (Fig 4D). In contrast, incubation of SAMHD1 with equivalent concentrations of dCTP could activate the dCTPase activity of SAMHD1, indicating that this concentration of dCTP

can induce oligomerisation (Appendix Fig S13A). In line with this, thermal shift assays of recombinant SAMHD1 revealed little difference in thermostability profiles between SAMHD1 incubated with either GTP and dCTPαS, or GTP and dATPαS (Appendix Fig S13B and C). Taken together, we propose a model in which inhibition of RNR leads to an imbalance of the dNTP pool, specifically an inversion of the dCTP-to-dATP ratio, causing a switch in the dNTP occupying the AS2 site that results in reduced ara-CTPase activity of SAMHD1 (Fig EV5H).

## Discussion

The deoxycytidine analogue ara-C remains the backbone treatment against AML (Mayer *et al*, 1994; Lowenberg, 2013). Clinical responses to ara-C correlate with accumulation of the active metabolite ara-CTP in AML cells (Plunkett *et al*, 1985), which is strongly regulated by the dNTPase SAMHD1 (Schneider *et al*, 2016; Herold *et al*, 2017a,b,c; Hollenbaugh *et al*, 2017; Rudd *et al*, 2017; Rassidakis *et al*, 2018). Thus, inactivation of SAMHD1 ara-CTPase is of immediate interest to rationally improve ara-C therapies. In this study, we embarked upon a phenotypic screening strategy to identify small molecules that could sensitise a SAMHD1-proficient AML cell line to ara-C but not their SAMHD1-deficient counterpart. We identified the deoxycytidine analogue dF-dC, clinically used to treat a range of solid malignancies (Toschi *et al*, 2005), as one such molecule. Subsequently, we show that the ability of this molecule to sensitise AML cells to ara-C in a SAMHD1-dependent manner is not through direct interaction with SAMHD1, but rather inhibition of its known target RNR. Accordingly, other RNRi such as HU and 3-AP, clinically used in AML (Mamez *et al*, 2016) and being evaluated in a number of clinical trials (Toschi *et al*, 2005), respectively, also displayed this phenomenon. We further demonstrate that these effects are restricted to non-allosteric RNRi inhibitors as known allosteric inhibitors Cl-F-ara-A, F-ara-A and 2-CdA did not display SAMHD1-dependent synergy with ara-C (see below for further details). The SAMHD1-dependent synergy for non-allosteric RNRi was observed in multiple cancer cell lines and patient-derived AML blasts and could be mechanistically linked to increasing intracellular ara-CTP concentrations, leading to induced DNA damage and apoptosis. These pharmacologic effects correlated with SAMHD1 protein abundance and, furthermore, could overcome the SAMHD1-mediated barrier to ara-C efficacy in AML xenograft mouse models. RNRi could lead to post-translational modification of SAMHD1: increased cysteine oxidation through ROS or altered T592 phosphorylation via Cdk1/2 inhibition. Indeed, oxidation can inactivate SAMHD1 (Mauney *et al*, 2017), and T592 phosphorylation has been reported to alter SAMHD1's substrate specificity (Jang *et al*, 2016). However, our experiments with a ROS scavenger and SAMHD1 phosphomutants did not implicate these modifications in RNRi-mediated ara-C sensitisation.

Small-molecule SAMHD1 inhibitors have been reported previously (Seamon *et al*, 2014; Seamon & Stivers, 2015; Hollenbaugh *et al*, 2017) (Appendix Table S1); however, whilst these molecules inhibit recombinant SAMHD1 *in vitro*, they have no demonstrated cell activity. More recently, a number of diverse FDA-approved drugs have been reported to inhibit hydrolysis of dGTP at micromolar concentrations *in vitro*, none of which inhibited dCTP hydrolysis,

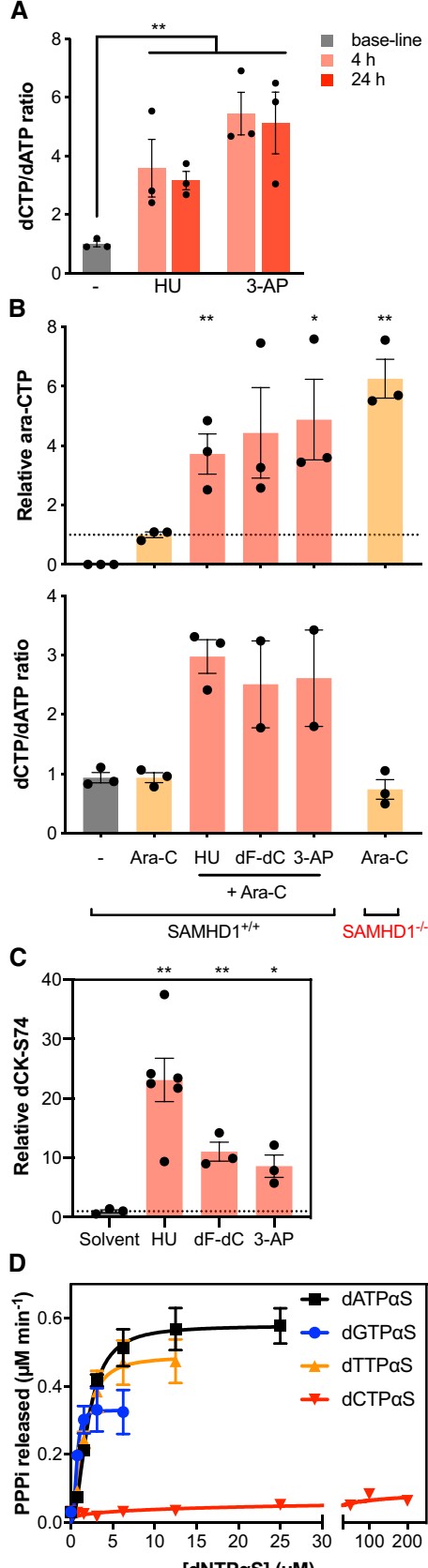

**Figure 4. dCTPαS-activated SAMHD1 is a poor ara-CTPase.**

A  Intracellular dNTP measurements using a primer extension assay in SAMHD1$^{+/+}$ THP-1 cells treated for 4 or 24 h with either 50 μM HU (middle panel) or 2.5 nM 3-AP (right panel), ratios of dCTP-to-dATP were calculated. Bars indicate mean values of three independent experiments; error bars indicate SEM. Statistical analyses were done using unpaired two-tailed *t*-tests: **$P$ < 0.01. For absolute values of dNTP pool measurements, see Appendix Fig S11.

B  Intracellular relative ara-CTP levels (upper panel) and dCTP:dATP ratio (lower panel) in the indicated cell lines following the indicated treatments determined using HPLC-MS/MS. SAMHD1$^{-/-}$ THP-1 cells were treated with 500 nM ara-C, and SAMHD1$^{+/+}$ THP-1 cells were treated with either solvent, 500 nM ara-C or a combination of 500 nM ara-C and an RNRi (HU, 50 μM; dF-dC, 10 nM; 3-AP, 150 nM) for 24 h. Values relative to mean ara-CTP amounts in ara-C-treated SAMHD1$^{+/+}$ THP-1 cells shown (indicated by dashed line). Circles, columns and error bars correspond to individual values, means and SEM of at least three experiments performed independently. Analyses were performed using unpaired two-tailed *t*-tests. *$P$ < 0.05; **$P$ < 0.01.

C  Quantification of dCK phosphorylation at serine-74 (S74) with respect to total dCK in SAMHD1$^{+/+}$ THP-1 cells treated with either solvent, 500 nM ara-C or a combination of 500 nM ara-C and an RNRi (HU, 50 μM; dF-dC, 10 nM; 3-AP, 150 nM) for 24 h. Circles and squares, columns and error bars correspond to individual measurements, means and SEM of one representative out of two independent experiments performed in triplicates. Statistical analyses were performed using unpaired two-tailed *t*-test. *$P$ < 0.05; **$P$ < 0.01.

D  Measurement of released inorganic triphosphate (PPPi) from hydrolysis of ara-CTP (200 μM) by recombinant SAMHD1 (0.35 μM) in the presence of GTP (200 μM) and a titration of different non-hydrolysable dNTP analogues (dNTPαS) in the enzyme-coupled malachite green assay. Error bars indicate SEM of two independent experiments performed in triplicate and quadruplet.

however (Mauney *et al*, 2018) (Appendix Table S1). Given the shortage of suitable small-molecule tools for cellular studies, we have alternatively proposed using Vpx as a biologic SAMHD1 inhibitor (Herold *et al*, 2017b). Also, this approach has limitations, as we discussed (Herold *et al*, 2017a). Importantly, here, we have demonstrated that drugs already in clinical use can be used to indirectly target SAMHD1 activity towards ara-CTP and thereby possibly overcome this barrier to ara-C efficacy in AML treatment. Critically, one of these, HU, is already used to treat AML, thus facilitating rapid translation of these findings to the clinic.

Synergy between RNRi and ara-C has been reported previously *in vivo* as well as in *ex vivo* experiments of primary patient blasts and in cell lines (Plagemann *et al*, 1978; Walsh *et al*, 1980; Streifel & Howell, 1981; Howell *et al*, 1982; Rauscher & Cadman, 1983; Tanaka *et al*, 1985; Kubota *et al*, 1988, 1989; Gandhi & Plunkett, 1990; Bhalla *et al*, 1991; Colly *et al*, 1992; Santini *et al*, 1996; Iwasaki *et al*, 1997; Freund *et al*, 1998; Heinemann *et al*, 1998; Ahlmann *et al*, 2001; Hubeek *et al*, 2004; Sigmond *et al*, 2007) (for further detail, see Appendix Table S3). Our study provides a mechanistic framework of this synergy in its dependence on functional SAMHD1 and expression levels thereof. Indeed, synergy of RNRi and ara-C, as well as an increase in intracellular ara-CTP levels, was predominantly reported in cell lines now known to be SAMHD1-positive and absent in SAMHD1-negative cell lines (Appendix Table S3), which is supported by the data presented here.

RNR is critical for the *de novo* production of dNTPs that in turn allosterically regulate SAMHD1 activity. This is particularly relevant for ara-CTP given this nucleotide species is not an allosteric

activator itself, and thus, ara-CTP hydrolysis is absolutely dependent upon the existing intracellular dNTP pool. We propose a model in which inhibition of RNR leads to a dNTP pool imbalance resulting in a switch in the dNTP occupying the AS2 site of SAMHD1, leading to reduced ara-CTPase activity (Fig EV5H). In support of this, firstly, we show that the direct effect of sub-toxic concentrations of RNRi in cultured cells causes significant imbalances of dNTP pools, rather than uniform depletion, in particular the inversion of the dCTP-to-dATP ratio. This is consistent with a previously reported fourfold to fivefold increased dCTP-to-dATP ratio in HU-treated cells (Julias & Pathak, 1998). Even though *in vitro* inhibition of RNR causes a uniform reduction of all four NDPs (Chimploy *et al*, 2000), inhibition of RNR in cells does not lead to uniform dNTP depletion, but rather to net pool imbalances. Specifically, purine dNTP pools are consistently depleted but pyrimidine dNTP pools are much less affected or even expanded (Snyder, 1984; Ahmad *et al*, 2017; Le *et al*, 2017). With regard to dCTP, whilst RNR inhibition results in a decrease in the *de novo* dCTP pool, generation of dCTP through salvage pathways can simultaneously increase, thus resulting in a net elevation of dCTP (Le *et al*, 2017). Accordingly, HU treatment has been reported to lead to increased activity of the salvage pathway enzymes thymidine kinase and dCK (Gao *et al*, 1995), and in support of this, here, we show that RNRi treatment increased phosphorylation of dCK at S74, known to lead to increased selectivity towards dC resulting in elevated dCTP salvage pools (Bunimovich *et al*, 2014; Le *et al*, 2017). DCK activation, which would increase the first intracellular phosphorylation step also for ara-C, might also explain additive effects of RNRi and ara-C observed in cells devoid of catalytically active SAMHD1. This is also consistent with a trend towards prolonged survival in the two SAMHD1-negative AML mouse models tested when comparing ara-C only with combined ara-C/HU treatment.

The three RNRi shown here to synergise with ara-C in a SAMHD1-dependent manner have distinct inhibitory mechanisms. HU scavenges the free tyrosyl radical in the active site of RRM2 and depletes the di-iron centre required for catalysis (Yarbro, 1992; Nyholm *et al*, 1993), whilst 3-AP forms a complex with $Fe^{2+}$ and interferes with the regeneration of RRM2 tyrosyl radical (Aye *et al*, 2012). In contrast, dF-dC is a suicide inhibitor, becoming covalently linked to the large RRM1 subunit (Wang *et al*, 2007). Interestingly, none of the tested purine nucleoside RNRi, which bind to the allosteric site on RRM1, synergised with ara-C in a SAMHD1-dependent manner. This discrepancy could be consistent with a time-dependent loss of RNR inhibitory activity of both Cl-F-ara-A di- and triphosphate (Aye & Stubbe, 2011) that is not observed for dF-dC diphosphate (Wang *et al*, 2007). More importantly, in contrast to non-allosteric HU and dF-dC (Gandhi *et al*, 1995; Smid *et al*, 2001; Guo *et al*, 2016), purine RNRi are reported to strongly reduce dCTP levels, and, consequently, dCTP-dATP ratios are not reversed (Sato *et al*, 1984; Griffig *et al*, 1989; Parker *et al*, 1991; Xie & Plunkett, 1996). However, further studies are needed to elucidate why allosteric purine nucleoside RNRi do not synergise with ara-C in a SAMHD1-dependent manner.

For cells expressing catalytically competent SAMHD1, we propose that the relative increase of dCTP, particularly in proportion to dATP, which is expected to typically occupy the AS2 site of SAMHD1 in unperturbed cells (Koharudin *et al*, 2014), could result in an increase of dCTP bound to this site. This shift might further be favoured by the fact that the affinity to AS2 is reported to be highest for dCTP, with a twofold, threefold and 10-fold lower apparent $K_m$ as compared to dGTP, dATP and dTTP, respectively (Jang *et al*, 2016). In addition, the lifetime of dCTP-induced tetramers is reported to be longer than the one of its dATP-induced counterparts (Wang *et al*, 2016). Secondly, in support of our model, we show that whilst allosteric activation of SAMHD1 with non-hydrolysable derivatives of dATP, dGTP and dTTP led to robust hydrolysis of ara-CTP *in vitro*, ara-CTPase activity is not detected when using the dCTP derivative, even though dCTP-activated SAMHD1 is clearly able to hydrolyse dCTP. This is consistent with the apparent lack of ara-CTPase activity in RNRi-treated cells and patient-derived AML blasts. In line with this, it has been previously demonstrated that dCTP as an AS2 activator can affect SAMHD1 substrate specificity (Jang *et al*, 2016; Wang *et al*, 2016). Nevertheless, future work will have to provide direct evidence for this model. That, under certain circumstances, SAMHD1 can have differential substrate specificity is illustrated by the fact that high SAMHD1 expression in macrophages leads to consistent reduction of dATP, dGTP, dCTP and dTTP, whilst dUTP levels remain high —even though dUTP is a strong allosteric activator of SAMHD1 itself (Kennedy *et al*, 2011; Hansen *et al*, 2014). Different allosteric activators can retain the overall structural properties of tetrameric SAMHD1, but subtle conformational changes can be induced; for example, the histidine-215 side chain in the catalytic site of SAMHD1 is positioned differently in GTP:dATP SAMHD1 as compared to dGTP:dATP SAMHD1 (Koharudin *et al*, 2014). Thus, future efforts to resolve the co-crystals of ara-CTP bound SAMHD1 with different allosteric activators could shed light on the phenomenon described here.

Intricate interplay between RNR and SAMHD1, both allosterically regulated by nucleotides and key enzymes in DNA precursor metabolism, is perhaps to be expected, and this will undoubtedly have relevance to the metabolism of nucleoside-based drugs. Consistent with that notion, certain *RRM1* single nucleotide polymorphisms (SNPs) were reported to be associated with reduced ara-CTP accumulation in ara-C-treated primary patient AML blasts, as well as worse survival (Cao *et al*, 2013). Furthermore, here we show that taking into account *RRM1*, *RRM2* or *RRM2B* expression in a multivariable model leads to increased HRs for SAMHD1.

Whilst the precise molecular mechanism by which dNTP imbalances resulting from RNR inhibition reduce SAMHD1 ara-CTPase activity remains to be fully elucidated, the implications of this phenomenon have immediate clinical impact. Critically, combinations of ara-C and non-allosteric RNRi for AML and other haematological malignancies could be implemented directly in current clinical practice, with SAMHD1 expression being a predictive biomarker for therapeutic efficacy. HU/ara-C combinations have been tested in early clinical trials with encouraging toxicity results (Sauer *et al*, 1976; Howell *et al*, 1982; Tanaka *et al*, 1985; Zittoun *et al*, 1985; Schilsky *et al*, 1987, 1992; Slapak *et al*, 1992; Frenette *et al*, 1995; Higashigawa *et al*, 1997; Yee *et al*, 2006; Dubowy *et al*, 2008; Odenike *et al*, 2008) (for further detail, see Appendix Table S4). Furthermore, HU is amongst the cheapest drugs used in oncology, and combinations with ara-C are thus not restricted to developed countries. Indeed, overcoming the SAMHD1-mediated barrier to ara-C efficacy that is responsible for worse OS in AML (Herold *et al*, 2017b; Rassidakis *et al*, 2018) is particularly

     

relevant for developing countries as they carry the major disease burden and death toll of AML (Ferlay et al, 2015).

Combination chemotherapy is the paradigm of systemic anti-cancer therapy, and effective combinations to date have largely been identified empirically. Our study presents a new mechanistic rationale for a combination treatment with cytotoxic drugs being superior to single-agent treatments. Put into a broader perspective, future work should systematically interrogate the underlying mechanistic basis for clinical efficacy of commonly used combination chemotherapies. Accordingly, combination chemotherapies could be tailored and become part of personalised precision medicine that hitherto has primarily focused on defined molecular targets (Chae et al, 2017).

# Materials and Methods

## Human cell lines

THP-1, HuT-78, in vivo adapted HL-60 (HL-60/iva) (Herold et al, 2017b) and their CRISPR/Cas9-generated derivatives, described previously (Herold et al, 2017b) or below, were cultured in Iscove's modified Dulbecco's medium (IMDM; GE Healthcare). KBM-7, K562, CCRF-CEM, MV-4-11, Jurkat and MOLT-4 were cultured in Roswell Park Memorial Institute medium (RPMI 1640 GlutaMAX; Thermo Fisher Scientific). All media were supplemented with 10% heat-inactivated foetal calf serum (Thermo Fisher Scientific) and 100 U/ml penicillin–100 μg/ml streptomycin (Thermo Fisher Scientific). Cell lines were purchased from ATCC except KBM-7 which were a gift from Dr. Nina Gustafsson (Kancera AB & Karolinska Institutet). All cell lines were regularly monitored and tested negative for the presence of mycoplasma using a commercial biochemical test (MycoAlert, Lonza). Cell line authentication was performed by Eurofins Genomics Europe Applied Genomics GmbH (Ebersberg, Germany) for luciferase-transduced SAMHD1-proficient and SAMHD1-deficient THP-1 cell clones. DNA isolation was carried out from cell pellet (cell layer). Genetic characteristics were determined by PCR-single-locus-technology. Sixteen independent PCR systems D8S1179, D21S11, D7S820, CSF1PO, D3S1358, TH01, D13S317, D16S539, D2S1338, AMEL, D5S818, FGA, D19S433, vWA, TPOX and D18S51 were investigated with proprietary primer sets. Cell lines were typically cultured in densities between $1–10 \times 10^5$ cells/ml at 37°C with 5% $CO_2$ in a humidified incubator.

## Generation of SAMHD1 CRISPR/Cas9 cell lines

Generation of THP-1 and HL-60/iva SAMHD1$^{+/+}$ and SAMHD1$^{-/-}$ cell clones was described previously (Herold et al, 2017b), referred to as THP-1 ctrl, THP-1 g2-2, HL-60/iva g2-3 and HL-60/iva g2-2, respectively (Herold et al, 2017b). Generation of firefly luciferase-expressing THP-1 SAMHD1$^{+/+}$ and SAMHD1$^{-/-}$ cell clones has been described (Herold et al, 2017c). Firefly luciferase-expressing HL-60 SAMHD1$^{+/+}$ and SAMHD1$^{-/-}$ cells were generated similarly by transducing HL-60 SAMHD1$^{+/+}$ clone g2-3 and SAMHD1$^{-/-}$ clone g2-2, respectively, with VSV-G pseudotyped lentiviral vector expressing HA-LUC (pCSXW-HALUC), previously described (Herold et al, 2017c). For reconstitution experiments, THP-1 SAMHD1$^{-/-}$ cell clone g2-2 was transduced with VSV-G pseudotyped lentiviral vector encoding gRNA-resistant SAMHD1 wild-type or catalytic-dead mutant H233A, as described before (Herold et al, 2017a). THP-1 SAMHD1$^{-/-}$ cell clone (g2-2) was transduced with VSV-G pseudotyped lentiviral vector encoding gRNA-resistant SAMHD1: either wild-type or the phosphomutants T592A and T592E, as described before (Herold et al, 2017a).

SAMHD1 CRISPR/Cas9 HuT-78 cell clones were generated by transducing HuT-78 cells with CRISPR/Cas9 lentiviral vector encoding gRNA g2 (Herold et al, 2017b). Cell bulks were selected with puromycin for 2 weeks, and single cell clones were generated by limiting dilution. As control, untransduced HuT-78 cell clones (SAMHD1$^{+/+}$) were generated in parallel.

Oligonucleotides encoding for SAMHD1 gRNAs were cac cgCTCGGGCTGTCATCGCAACG (fwd g2) and
aaacCGTTGCGATGACAGCCCGAGc (rev g2), as well as cacc gATCGCAACGGGGACGCTTGG (fwd g3)
and aaacCCAAGCGTCCCCGTTGCGATc (rev g3).

## Production of VLPs

SIV$_{MAC}$ VLP production, either control or packaged with Vpx, was described previously (Herold et al, 2017b) and references therein.

## Primary AML blasts

Experiments with primary paediatric and adult AML blasts were approved by the regional ethical review board in Stockholm (no. 03-810, no. 02-445, no. 2013/1248-31/4 and no. 2013/1248-31/4), and informed consent was obtained. Experiments conformed to the principles set out in the WMA Declaration of Helsinki and the Department of Health and Human Services Belmont Report. Clinical and cytogenetic parameters can be found in Appendix Table S2. Cells were thawed and cultured at a density of $1 \times 10^6$ cells per ml in filtered StemPro-34 SFM medium with StemPro Nutrient Supplement (cat no. 10639011; Thermo Fisher Scientific), containing 10% foetal bovine serum (FBS; GE Healthcare), 100 U/ml penicillin and 100 μg/ml streptomycin (cat no. 15070063; Thermo Fisher Scientific). The medium was further supplemented with the following recombinant cytokines in a concentration of 20 ng/ml: IL-6 (cat no. 206-IL-010), IL-3 (cat no. 203-IL-010; both R&D Systems), TPO (cat no. 02822) and GM-CSF (cat no. 78015.1; both Stemcell Technologies).

For Vpx treatment, 24 h after thawing, $10 \times 10^6$ cells were collected, spun down and resuspended in 2 ml medium. Cells were equally distributed in 10 wells of a 24-well plate and were treated with either 50 μl of Vpx-VLPs or 50 μl of control VLPs each (two groups of five wells, $1 \times 10^6$ cells total for each treatment). Cells were incubated for 3 h at 37°C prior to collection and pooling before an additional 8.5 ml of medium was added to increase the final volume to 10 ml and cell density of $0.5 \times 10^6$ per ml. Cells were incubated overnight prior to further processing.

## Compound preparation

Cytarabine (ara-C) was purchased from Jena Bioscience, Germany (cat no. N-20307-5), and Sigma-Aldrich, Sweden (cat no. C1768), gemcitabine (dF-dC; cat no. G6423), hydroxyurea (HU; cat no. H8627), triapine (3-AP; cat no. SML0568), clofarabine (Cl-F-ara-A; cat no. C7495), fludarabine (2-F-ara-A; cat no. F2773) and

cladribine (2CdA; cat no. C4438), were purchased from Sigma-Aldrich, Sweden, and MK-1775 (cat no. SC-483196) was purchased from Santa Cruz Biotechnology, USA. Compounds were typically prepared as 10–50 mM stock solutions in DMSO and were stored at −20°C, with the exception of HU which was prepared fresh. Alternatively, when not being used with liquid handling equipment, ara-C and HU were prepared in water; no difference in $EC_{50}$ was observed between DMSO and water stocks. N-acetyl cysteine (NAC; Sigma-Aldrich) was dissolved in 1 M HEPES solution (Thermo Fisher Scientific) to a concentration of 0.5 M and pH adjusted to 7.2. Prior to use, this stock solution was diluted to 5 mM in complete cell media.

**Phenotypic screen**

For the screen, the following compound libraries were used: Scilifelab Primary Screening set (30K), SelleckChem known tool cpds: L1700, Tocris mini known tool cpds: #2890, Prestwick chemical lib: PCL-1200. Assay plates were prepared by transferring 30 nl of 10 mM DMSO compound solutions and controls using acoustic dispensing (Echo 550, Labcyte) to white 384-well assay plates (Corning 3570). Compounds were placed in columns 1–22. Thirty nanoliter DMSO (negative control) was placed in column 23, and 30 nl 205 mM Ara-C (positive control) was placed in column 24. The plates were then heat-sealed using peelable aluminium seal (Eppendorf, 0030127790) with a thermal microplate sealer (PlateLoc, Agilent) and then stored at −20°C until use. On the day of the experiment, the plates were allowed to thaw for 30 min followed by a brief centrifugation step (1,000 g for 1 min) prior to removal of the seal. The final compound concentration in the screen was 10 μM, and the final DMSO concentration was 0.1%. The final concentration of the positive control was 205 μM ara-C.

THP-1 SAMHD1$^{+/+}$ cells were diluted with cell culture medium. For experiments in the presence of ara-C at $EC_{10}$, cells were treated with ara-C to a final concentration of 400 nM. Next, 1,000 cells per well were dispensed using a MultiDrop device (Thermo Fisher Scientific) to the assay plates already containing the test compounds; the final volume in the assay plates was 30 μl. The plates with cells were placed in a plastic container with damp cloths to create a humid atmosphere. The box was incubated for 72 h at 37°C, 5% $CO_2$ in a humidified incubator. The plates were removed from the incubator and were placed at room temperature for ~ 30 min to allow equilibration to room temperature. Next, 30 μl CellTiter-Glo® Luminescent Cell Viability Assay (G7573; Promega) diluted with an equal volume of water was added to the plates. The plates were placed on an orbital shaker for ~ 3 min and were then incubated for at least 7 min prior to reading the luminesce using an Envision plate reader (PerkinElmer).

**Proliferation inhibition assays and drug synergy analysis**

Compound dilution series in DMSO were dispensed into 384-well plates using either an Echo® 550 Liquid Handler (Labcyte) or a D300e Digital Dispenser (Tecan). The DMSO volume was normalised across the plate, not exceeding a total volume of 500 nl per well. Shortly after, cells (1,000 cells per well in 50 μl medium) were dispensed into these plates using a MultiDrop (Thermo Fisher Scientific); typically, to facilitate cell dispensing,

FCS in cell medium was reduced to 5%. Plates were incubated in a humidified chamber at 37°C with 5% $CO_2$ for 72–96 h before addition of 10 μl resazurin (Sigma-Aldrich, Sweden; cat no. R7017; 0.06 mg/ml in PBS) and were further incubated for 6 h before measurement of fluorescence at 530/590 nm (ex/em) using a Hidex Sense Microplate Reader. Fluorescence intensity for each well was normalised to the average of control wells on the same plate containing cells with DMSO (100% viability control) and medium with DMSO (0% viability control). The data were analysed using a four-parameter logistic model in Prism 7 (GraphPad Software).

The proliferation inhibition assay used for the synergy study on primary AML blasts and the small-molecule screen was performed using the ATP-release assay CellTiter-Glo® (cat no. G7573, Promega) instead of the resazurin-based assay described above. The experimental setup was similar to the resazurin assay with a few exceptions. The dispensed volume of cells was 30 μl per well containing 1,000 cells for experiments with cell lines and 15,000 cells for experiments using primary AML blasts (treated according to primary AML blasts section). For the experiments with sub-lethal dose of ara-C, the cells were treated with 266 or 400 nM ara-C prior to dispensing the cells. After 72-h incubation as described above, the plates were removed from the incubator and placed at room temperature for ~ 30 min to allow equilibration to room temperature. Thirty microliter CellTiter-Glo® diluted with an equal volume of water was added to the plates using a MultiDrop. The plates were placed on an orbital shaker for ~ 3 min and were then incubated for at least 7 min prior to reading the luminesce using an Envision plate reader (PerkinElmer).

For synergy experiments, compound dispensing was performed exclusively with the Tecan D300e Digital Dispenser using the Synergy Wizard in the D300e Control Software. Prior to these experiments, single-compound concentration–response curves were performed to determine the concentration range to be used in the concentration–response matrix, ideally choosing a dilution series to obtain a complete concentration–response curve with each compound individually. Cells were added to plates containing compound dilution series, and incubated for 72 h before measurement of cell viability, with the exception of the experiment performed in the cell line panel, which were incubated for 96 h. The average relative cell viability measurement from duplicate wells for the dose–response matrix was compiled into a data frame for analysis in the R-package Synergyfinder (Yadav et al, 2015). A dose–response landscape using the ZIP or HSA models was generated and an average synergy score across the landscape calculated. The HSA model defines synergy as a combinatorial effect that is larger than the individual drug effect observed at the same concentration, whilst the ZIP model combines the widely used Bliss independence and Loewe additivity models into a response surface model that uses a delta score to characterise synergy. The delta score, derived from the ZIP method, denotes the percentage of proliferation inhibition observed over the expected response, with a score of 0, > 0 or < 0 corresponding to zero interaction, synergy or antagonism, respectively. Based upon the original study (Yadav et al, 2015), a delta score > 5 was categorised as strong synergy, whilst < 5 was categorised as strong antagonism. The excess over HSA score denotes the sum of differences between the combination effect and the expected highest single-agent effect.

## Orthotopic AML mouse models using human AML cell lines

All animal experiments were carried out as per guidelines from Swedish animal welfare rules and regulations as stated by the Swedish Board of Agriculture. Experimental protocols were approved by the regional animal ethical committee in Stockholm in compliance with EU directive 2010/63 and followed the guidelines stated in ethical applications #N89/14 and 5718-2019. Mice were housed in required controlled environmental condition with food and water *ad libitum*. Sample sizes for animal studies were based on our experiences (Herold *et al*, 2017b) and were estimated to be 5–6 per group with a power of 0.8 and a significance level of 0.05, estimating a hypothetical difference in median survival of 20 days with an SD of 12 days upon successful intervention. To make THP-1 and HL-60/iva orthotopic models, 5 million cells (either THP-1 or HL-60/iva SAMHD1$^{+/+}$ and SAMHD1$^{-/-}$ cell clones expressing firefly luciferase) in PBS were injected into NOD/SCID IL2R$^{-/-}$ female mice by tail vein injection. Later, mice were randomly divided into four different groups: vehicle, ara-C, HU and combination of ara-C and HU. Six days after cell injection, vehicle or drugs (either alone or in combination) were injected by intraperitoneal injections into the mice, once a day for five consecutive days (dose for THP-1 study: vehicle—NS, ara-C—100 mg/kg, HU—75 mg/kg and ara-C—100 mg/kg + HU—75 mg/kg; for HL-60/iva study: vehicle—NS, ara-C—50 mg/kg, HU—75 mg/kg and ara-C—50 mg/kg + HU—75 mg/kg). An additional study was performed comparing the effects of vehicle, ara-C, dF-dC and combination of ara-C and dF-dC in the THP-1 model. dF-dC was only administered on days 1 and 3 of the 5-day treatment, both as a single treatment or in combination with ara-C (in the latter case, ara-C was given alone on days 2, 4 and 5). Doses: vehicle—NS, ara-C—100 mg/kg, dF-dC—20 mg/kg and ara-C—100 mg/kg + HU —20 mg/kg. Tumour progression and metastasis were monitored using bioluminescence IVIS imaging system using Caliper Spectrum CT. Before taking images, 10 μl/g (ᴅ Luciferin sodium salt 15 mg/ml in PBS; cat no. L9504; Sigma-Aldrich, USA) was injected into the mice by intraperitoneal injection. High photon counts, external multiple tumours, single big tumour, more than 15% body weight loss or appeared sick was considered as study end-point. If animals died from apparent treatment toxicity more than 2 weeks before onset of leukaemic symptoms in the control group, animals were censored. Pathological gross examination of lymph node tumour and abnormalities in lung, liver, spleen, kidney, if any, was recorded during autopsy.

## Orthotopic immunocompetent murine AML mouse model

All mice were maintained in a specific pathogen-free condition in the animal facility of Karolinska Institutet. Animal procedures were performed with approval from the local ethics committee (ethical number 1869). Twelve- to fourteen-week-old wild-type CD45.2 C57BL/6J mice carrying the CD45.2 antigen in their leucocytes were used for the transplantation of *MLL-AF9* retrovirally transduced CD45.1$^{+}$ mouse AML cells. AML cells were generated from a syngeneic CD45.2 C57BL/6J mouse as previously described (Xiao *et al*, 2018). *MLL-AF9* AML cells were expanded in culture in the presence of IL-3 (5 ng/ml, R&D Systems) in RPMI + 10% FBS, and 250,000 *MLL-AF9*-expressing AML cells were

intravenously transplanted into the non-irradiated CD45.2 C57BL/6J mice. At 20 days post-AML cell injection, Ara-C (75 mg/kg; Sigma-Aldrich) and hydroxyurea (HU, 75 mg/kg; Jena Bioscience) either alone or in combination (Ara-C 75 mg/kg + HU 75 mg/kg) were injected intraperitoneally once a day for five consecutive days. The control group of the mice were injected with NS. The AML progression and engraftment were monitored by analysis of peripheral blood using a haematology analyser (Sysmex-XP300) and flow cytometry. The onset of AML and survival rate of the mice were assessed based on the general health condition (such as slow movement, hunch back, paralysed leg) of the mice after treatment.

## Cellular thermal shift assay (CETSA)

Cells were treated with compounds at a density of $0.5 \times 10^6$ cells/ml for the indicated time before collection and washing in PBS. Cell pellets were resuspended in Tris-buffered saline (TBS) supplemented with protease inhibitor cocktail (cOmplete™, Mini, EDTA-free Protease Inhibitor Cocktail; cat no. 04693159001; Roche), 60 μl per $1 \times 10^6$ cells, and aliquoted in PCR strip tubes ($1 \times 10^6$ cells per tube). Samples were heated for 3 min at the indicated temperature ranging from 38 to 60°C, followed by a 3-min incubation at room temperature. Cells were lysed in three freeze-thawing cycles consisting of a 3-min ethanol and dry ice bath, followed by a 3-min incubation at 37°C in a water bath; samples were vigorously mixed after each cycle by vortexing. Lysates were centrifuged for 20 min at 17,000 *g* to pellet the denatured and aggregated protein, and 45 μl of the supernatant was transferred to a new tube and 15 μl 4× Laemmli Sample Buffer (Bio-Rad) supplemented with 100 mM DTT was added prior to boiling. Following Western blot analysis, band intensities were quantified using Image Studio Lite software (LI-COR Bioscience) and normalised to a thermostable protein loading control (either SOD-1 (Miettinen & Bjorklund, 2014) or NUDT5 (Page *et al*, 2018)) before plotting and curve fitting (Boltzmann sigmoidal) using Prism 7 (GraphPad Software) to determine the $T_{agg}$.

## *In situ* chemical cross-linking

Cells at a density of $0.5 \times 10^6$ cells/ml were treated with compounds for the indicated time before collection by centrifugation, washing in PBS, aliquoting into 1.5-ml tubes ($1 \times 10^6$ cells per tube) and pelleting by centrifugation. Chemical cross-linker disuccinimidyl glutarate (DSG; cat no. 20593, Thermo Fisher Scientific) was prepared fresh in anhydrite DMSO (cat no. 1029310161, Merck Millipore) to a stock concentration of 25 mM. DSG stock was diluted in PBS to the desired concentration (5–0.3 mM) and each cell pellet resuspended in 50 μl followed by a 30-min incubation at room temperature. The reaction was quenched with the addition of 1 ml 1M Tris–HCl pH 8, and samples were incubated for a further 30 min at room temperature before collection by centrifugation (800 g for 5 min) and processing for Western blot analysis.

## Primer extension assay for measurement of intracellular dNTPs

Cells at a density of $0.5 \times 10^6$ cells/ml were treated with compounds for the indicated time prior to collection and PBS

washing. Cellular dNTP levels were measured by HIV-1 RT-based dNTP assay (Diamond *et al*, 2004). Briefly, cellular dNTPs were extracted from cells by 60% methanol and dried. The dried dNTP samples were blinded prior to resuspension and direct application to the RT-based primer extension reaction, which determines the amounts of dNTPs in the extracted samples. The dNTP amounts were normalised by $1 \times 10^6$ cells. Four different 19-mer DNA templates containing sequence variations (N) at the 5′ end nucleotide (5′-NTGGCGCCCGAACAGGGAC-3′) were individually annealed to an 18-mer DNA primer (5′-GTCCCTGTTCGGGCGCCA-3′), which was $^{32}$P-labelled at its 5′ end (template: primer, 4:1). The nucleotide at the 5′ end of the primer determines the dNTP to be measured.

## HPLC-MS/MS assay for measurement of intracellular dNTPs and ara-CTP

To simultaneously quantify the intracellular dNTPs and ara-CTP, an ion pair chromatography–tandem mass spectrometry method (Fromentin *et al*, 2010) was applied, with modifications. HPLC separation and MS detection were performed on a Vanquish Flex system (Thermo Scientific, Waltham, MA) coupled with a TSQ Quantiva triple quadrupole mass spectrometer (Thermo Scientific, Waltham, MA). Analytes were separated using a Kinetex XB-C18 column (150 × 2.1 mm, 2.6 μm; Phenomenex, Torrance, CA) at a flow rate of 250 μl/min, 40°C. Mobile phase A consisted of 2 mM of ammonium phosphate monobasic and 3 mM of hexylamine and mobile phase B consisted of acetonitrile. The LC gradient increased from 5 to 10% of mobile phase B in 15 min, 10 to 40% in 4 min and then returned to the initial condition in 0.5 min. Selected reaction monitoring in both positive and negative modes (spray voltage: 3,200 V [pos] or 2,500 V [neg]; sheath gas: 35 Arb; auxiliary gas: 20 Arb; ion transfer tube temperature: 350°C; vaporiser temperature: 380°C) was used to detect the targets: dATP (492 → 136, pos), dGTP (508 → 152, pos), dCTP (466 → 158.9, neg), TTP (481 → 158.9, neg) and ara-CTP (484 → 112, pos). Extracted samples were reconstituted in 400 μl of mobile phase A. After filtered through 0.2 μm membrane, 20 μl of filtrate was mixed with 10 μl of $^{13}$C- and $^{15}$N-labelled dNTPs as internal standards and further diluted 5 times before subjected to LC-MS analysis. Injection volume was 5 μl. Data were collected and processed by Thermo Xcalibur 3.0 software. Calibration curves were generated from standards by serial dilutions in mobile phase A (dNTPs and ara-CTP 1.25–1,000 nM). The calibration curves had $r^2$ value > 0.99. All the chemicals and standards are analytical grade or higher and were obtained commercially from Sigma-Aldrich (St. Louis, MO). Nucleotides were at least 98% pure.

## Western blot analysis

Sample preparation, SDS–PAGE and Western blot analysis were performed as described previously (Drakos *et al*, 2007; Herold *et al*, 2017b). The following primary antibodies were used in this study: SAMHD1 (Bethyl, A303-691A, 1:2,000; Abcam, ab128107, 1:1,000; Proteintech Group, 12586-1-AP, 1:1,000), SOD-1 (Santa Cruz, sc-11407, 1:3,000), RRM1 (Proteintech Group, 60073-2-1G, 1:1,000), RRM2 (Sigma-Aldrich, WH0006241M1, 1:1,000), RRM2B (Abcam, ab8105, 1:1,000), NUDT5 (in-house (Page *et al*, 2018), 1:1,000), Chk1-pS345 (Cell Signaling, 2341, 1:750), Chk2-pT68 (Cell Signaling, 2661, 1:750), Cleaved-PARP (Cell Signaling, 9541, 1:1,000),

γH2A.x (Millipore, 05-636, 1:2,000) and β-actin (Abcam, ab6276, 1:5,000; Santa Cruz, sc-47778 HRP, 1:2,000).

## Enzyme-coupled SAMHD1 activity assay

Production of recombinant human SAMHD1 and *Escherichia coli* inorganic pyrophosphatase (PPase), together with the assay method, was described previously (Herold *et al*, 2017b). All (d) NTPs and their analogues were purchased from Jena Biosciences.

## Differential scanning fluorimetry

Differential scanning fluorimetry was performed as described before (Valerie *et al*, 2016). Briefly, recombinant SAMHD1 protein (5 μM), Sypro Orange (5X; Thermo Fisher Scientific), and DMSO or nucleotides of various concentrations were combined in assay buffer (12.5 mM Tris-acetate, pH 7.5, 20 mM NaCl, 0.5 mM MgAc, 0.25 TCEP) in 96-well PCR plates at the final volume of 20 μl/well and DMSO concentration of 1%. The assay mixture was then subject to a 25–95°C temperature gradient with fluorescence intensities measured every minute, on a LightCycler 480 Instrument II (Roche). Melting temperatures were determined by LightCycler 480 Software.

## Statistical methods

The distributions of mRNA levels of *SAMHD1*, *RRM1*, *RRM2* and *RRM2B* in the TARGET and TCGA AML patient cohorts were evaluated using histograms and normal qq-plots. A natural log transformation was used for all four variables. Cox proportional hazards models were used to estimate HR for death (OS) or event (EFS), along with *P*-values and 95% confidence intervals. The proportional hazards assumption (PHA) was assessed by plotting Schoenfeld residuals against (a log transformation of) time and testing deviations from a zero slope. No deviation from the PHA was detected. Statistical analysis was carried out using R version 3.3.1 (R Foundation for Statistical Computing, Vienna, Austria).

Pearson and Spearman correlations, Kaplan–Meier survival analyses using Mantel–Cox log-rank test for animal studies, as well as unpaired two-sided *t*-tests and ANOVAs, were all performed using Prism 7 (GraphPad Software). Statistical analyses for relevant experiments were performed using Prism 7 software (GraphPad). Statistical parameters for main figures are listed below; for supplemental figures, see legends. For comparison of drug synergy scores, statistical significance was determined using a two-tailed unpaired *t*-test (Fig 1E—for HU: THP-1 $^{+/+}$ vs $^{-/-}$, $n = 7$, $P = < 0.0001$, $t = 6.09$, $df = 12$; THP-1 WT vs H233A, $n = 6$, $P = < 0.0001$, $t = 7.711$, $df = 10$; HuT-78 $^{+/+}$ vs $^{-/-}$, $n = 6$, $P = 0.0140$, $t = 2.972$, $df = 10$; HL-60 $^{+/+}$ vs $^{-/-}$, $n = 7$, $P = 0.0038$, $t = 3.577$, $df = 12$. For dF-dC: THP-1 $^{+/+}$ vs $^{-/-}$, $n = 6$, $P = 0.002$, $t = 4.065$, $df = 10$; THP-1 WT vs H233A, $n = 5$, $P = 0.0015$, $t = 4.707$, $df = 8$; HuT-78 $^{+/+}$ vs $^{-/-}$, $n = 5$, $P = 0.0130$, $t = 3.178$, $df = 8$; HL-60 $^{+/+}$ vs $^{-/-}$, $n = 5$, $P = 0.0379$, $t = 2.484$, $df = 8$. For 3-AP: THP-1 $^{+/+}$ vs $^{-/-}$, $n = 4$, $P = 0.0004$, $t = 7.208$, $df = 6$; THP-1 WT vs H233A, $n = 4$, $P = < 0.0001$, $t = 13.53$, $df = 6$; HuT-78 $^{+/+}$ vs $^{-/-}$, $n = 4$ and 3, respectively, $P = 0.0007$, $t = 7.468$, $df = 5$; HL-60 $^{+/+}$ vs $^{-/-}$, $n = 4$, $P = 0.3754$, $t = 0.9573$, $df = 6$). For Kaplan–Meier analyses, statistical significance was determined using Mantel–Cox log-rank test (Fig 2A—for SAMHD1 $^{+/+}$: vehicle vs HU, $n = 5$ and 6, respec-

## The Paper Explained

### Problem

The nucleoside analogue cytarabine (ara-C) is a cornerstone in the treatment of acute myeloid leukaemia (AML), but resistance to this drug will result in therapy failure and death. SAMHD1 is a dNTP triphosphohydrolase that chemically inactivates the active triphosphate metabolite of ara-C. Thus, targeting SAMHD1 to enhance ara-C efficacy is a rational strategy to improve survival in AML and other haematological malignancies. However, to date, there are no clinically viable strategies to achieve this.

### Results

Using an unbiased phenotypic screening strategy, we identified clinically used anti-cancer drugs—inhibitors of the nucleotide biosynthetic enzyme ribonucleotide reductase (RNRi)—that can be used to indirectly target the ara-CTP hydrolytic activity of SAMHD1. In various AML models, including cultured cell lines, patient-derived AML blasts and AML mouse models, we demonstrate that RNRi synergise with ara-C in a manner that is dependent upon SAMHD1. Using biophysical and biochemical methods, we propose a model in which nucleotide pool imbalance resulting from inhibition of RNR perturbs the allosteric activation of SAMHD1, resulting in a reduction of ara-CTP hydrolysis.

### Impact

Combining ara-C with an RNRi promises to overcome SAMHD1-mediated drug resistance in AML and other haematological malignancies. At least two non-allosteric RNRi are FDA- and EMA-approved and are currently employed in cancer treatment. In particular, hydroxyurea is an excellent candidate as it is affordable, as patents have expired, and an indication in AML already exists. Thus, it is a prime candidate for direct translation of our findings into clinical practice, which can be achieved by adding moderate doses to each administration of ara-C when treating AML.

---

tively, $P = 0.9486$, $\chi^2 = 0.004149$, $df = 1$; vehicle vs ara-C, $n = 5$ and 6, respectively, $P = 0.3173$, $\chi^2 = 1$, $df = 1$; vehicle vs ara-C + HU, $n = 5$ and 3, respectively, $P = 0.0082$, $\chi^2 = 7$, $df = 1$; HU vs ara-C + HU, $n = 6$ and 3, respectively, $P = 0.0079$, $\chi^2 = 7.059$, $df = 1$; ara-C vs ara-C + HU, $n = 5$ and 3, respectively, $P = 0.0141$, $\chi^2 = 6.028$, $df = 1$. For SAMHD1$^{-/-}$: vehicle vs ara-C, $n = 6$, $P = 0.0012$, $\chi^2 = 10.48$, $df = 1$. Fig 2B—for SAMHD1$^{+/+}$: vehicle vs HU, $n = 6$, $P = 0.5999$, $\chi^2 = 0.2752$, $df = 1$; vehicle vs ara-C, $n = 6$, $P = 0.1845$, $\chi^2 = 1.761$, $df = 1$; vehicle vs ara-C + HU, $n = 6$, $P = 0.0186$, $\chi^2 = 5.542$, $df = 1$; HU vs ara-C + HU, $n = 6$, $P = 0.0220$, $\chi^2 = 5.248$, $df = 1$; ara-C vs ara-C + HU, $n = 6$, $P = 0.0316$, $\chi^2 = 4.619$, $df = 1$. For SAMHD1$^{-/-}$: vehicle vs ara-C, $n = 6$, $P = 0.0012$, $\chi^2 = 10.56$, $df = 1$. Fig 2C—vehicle vs ara-C: $n = 7$, $P = 0.6535$, $\chi^2 = 0.2016$, $df = 1$; vehicle vs dF-dC: $n = 7$ and 6, respectively, $P = 0.0682$, $\chi^2 = 3.326$, $df = 1$; vehicle vs ara-C + dF-dC: $n = 7$ and 6, respectively, $P = 0.0011$, $\chi^2 = 10.64$, $df = 1$; ara-C vs ara-C + dF-dC: $n = 7$ and 6, respectively, $P = 0.0014$, $\chi^2 = 10.22$, $df = 1$; dF-dC vs ara-C + dF-dC: $n = 6$, $P = 0.0097$, $\chi^2 = 6.685$, $df = 1$. Fig 2D—vehicle vs ara-C: $n = 8$ and 5, respectively, $P = 0.0995$, $\chi^2 = 2.713$, $df = 1$; vehicle vs HU: $n = 8$ and 5, respectively, $P = 0.2184$, $\chi^2 = 1.515$, $df = 1$; vehicle vs ara-C + HU: $n = 8$ and 5, respectively, $P = 0.0026$, $\chi^2 = 9.075$, $df = 1$). Statistical testing of paired drug synergy plots was determined using two-way ANOVA (Fig 3G—$n = 12$, $F = 12.6$, $df = 1$).

For statistical testing of dNTP pool ratios, unpaired two-tailed $t$-tests were used (Fig 4A—HU vs. untreated: $t = 11.27$; $df = 2$; $P = 0.0078$. 3-AP vs. untreated: $t = 26.72$; $df = 2$; $P = 0.0014$). For statistical testing of ara-CTP levels, unpaired two-tailed $t$-tests were used (Fig 4B—ara-C vs ara-C + HU: $n = 3$, $P = 0.0162$, $t = 3.994$, $df = 4$; ara-C vs ara-C + dF-dC: $n = 3$, $P = 0.0880$, $t = 2.246$, $df = 4$; ara-C vs ara-C + 3-AP: $n = 3$, $P = 0.0463$, $t = 2.852$, $df = 4$; ara-C vs ara-C in THP-1 SAMHD1-/-: $n = 3$, $P = 0.0014$, $t = 7.950$, $df = 4$). For statistical testing of dCK-S74 abundance, unpaired two-tailed $t$-tests were used (Fig 4C—solvent vs HU: $n = 3$ and 6, respectively, $P = 0.0043$, $t = 4.141$, $df = 7$; solvent vs dF-dC: $n = 3$, $P = 0.0034$, $t = 6.198$, $df = 4$; solvent vs 3-AP: $n = 3$, $P = 0.0160$, $t = 4.009$, $df = 4$).

**Expanded View** for this article is available online.

## Acknowledgements

We are grateful to A. Björklund and S. Bengtzén for assistance with primary AML samples. We acknowledge D. Gavhed for administrative assistance. We would like to express our gratitude to NCI's Office of Cancer Genomics (OCG), The Cancer Genome Atlas (TCGA) and Therapeutically Applicable Research To Generate Effective Treatments (TARGET). This work was supported by grants from the Swedish Children's Cancer Foundation (TJ2017-0021 (to S.G.R.); FR2017-0154 (to H.Q.) PR2017-0113 (to K.P.T.); PROF06/001, PR2015/005 and KP2018-0005 (to J.-I.H); PR2013-0002 and PR2014-0048 (to T.H.); TJ2016-0040, PR2016-0044 and PR2018-0016 (to N.H.)), the Swedish Cancer Society (19-0056-JIA to S.G.R.; CAN 2017/774 to H.Q.; CAN 2014/814 to Sö.L; CAN 2012/770 and CAN 2015/255 to T.H.; CAN 2013/396 and CAN 2016/275 to J.-I.H.; and CAN 2017/517 to N.H.), the Swedish Medical Association (SLS-875361 to N.H.), the Clas Groschinsky Memorial Foundation (M18228 to N.H.), the Mary Béve Foundation for childhood cancer research (to N.H.), the Harald och Greta Jeanssons Foundation (2018 and 2019 to N.H.), the Åke Wiberg Foundation (M18-0012 to N.H.), the Lars Hierta Memorial Foundation (FO2018-0002 to N.H.); the ìShizu Matsumuraî donation (2018-01086 to N.H.), the Sigurd och Elsa Goljes Memorial Foundation (LA2018-0038 to N.H.), the Swedish Research Council (2018-02114 to S.G.R.; 2015-02498 to Sö.L.; 2011-3897 to J.-I.H., and 2012-5935, 2013-3791 and 2015-00162 to T.H.), Radiumhemmet's Research Foundations (171162 to H.Q.; 154242 to G.R.; 174122 to D.G. and 191112 to N.H.), the Torsten and Ragnar Söderberg Foundation (to T.H.), the Stockholm County Council (ALF; 20150353 to Sö.L.; 20150016 and 20180318 to J.-I.H.; and K2892-2016 to N.H) the Alex and Eva Wallström Foundation for scientific research and education (2017-00475 and 2018-00109 to S.G.R.), the Felix Mindus contribution to Leukemia Research (2016-52575, 2017-01287, 2018-02716 to S.G.R.; 2018-02715 to N.H.; 2019-02004 to S.M.Z.), The Loo and Hans Osterman Foundation for Medical Research (2019-01128 to S.G.R.), the Märta and Gunnar V Philipsons Foundation (to J.-I.H. and N.H.) and Karolinska Institutet Foundations (2016-50273 & 2018-01573 to S.G.R.; and 2016-50756 & 2018-01086 to N.H.). This work was further supported by the National Institute of Health (AI136581 and GM104198 to B.K.; MH116695 to R.F.S.). J.K. was supported by the German Research Foundation (SCHA1950/1-1). Chemical Biology Consortium Sweden is funded by the Swedish Research Council, Science for Life Laboratories and Karolinska Institutet (829-2009-6241 to H.A. and T.L).

## Author contributions

The overall study was conceived by SGR and NH. The manuscript was written by SGR and NH, revised by CBJP, NT, HA, TL, J-IH, TH and TS, and read by all authors. Data and figure assembly were performed by SGR and NH

and viewed by all authors. The project was supervised by SGR and NH. CRISPR/Cas9 knockout cell lines and Vpx-VLPs were generated by JK, SSB and TS Small-molecule screening strategy was conceived by NH, discussed with TL and performed by HA. Concentration matrix drug synergy assays were designed by SGR, CBJP and NH, method established by SGR and CBJP, and subsequent experiments were performed by SGR, NT, CBJP, SMZ and PM. Compound handling was performed by SGR, CBJP and EW. Cellular thermal shift assays were designed by SGR and NH, method established by SGR, NT and SL and subsequent experiments performed by NT and SGR. Chemical cross-linking experiments were designed by SGR and NH and performed by NT and SGR. Animal experiments were designed by SGR, KS, UW-B, JW, HQ and NH and performed by KS, LS, SS, AR and HQ. DNA damage signalling experiments were designed by SGR and performed by SGR and NT. Experiments with primary patient-derived AML blasts were designed and performed by SGR, NT, CBJP, NT, SöL, MH, DG, KPT, GR, HA and NH. *In vitro* biochemical assays were designed by SGR, CBJP and NH and performed SGR and CBJP. *In vitro* biophysical assays were designed by SGR, SMZ and NH and performed by SMZ. Nucleotide pool measurement experiments were designed by NH and SGR, samples prepared by SGR and NT, and subsequent analysis performed by S'AC and ST under the supervision of BK and RFS. Experiments of dCK phosphorylation were planned by RMB, JL and NH; samples were prepared by NT; and analyses were carried out by GM and RMB. Analysis of TCGA and TARGET data was performed by IHM and discussed with J-IH and NH. Funding for the project was acquired by SGR, J-IH, TH and NH.

## Conflict of interest

T.L. is employed at AstraZeneca. C.B.J.P is currently employed by Research Institutes of Sweden.

## For more information

(i) https://staff.ki.se/people/searud

(ii) https://staff.ki.se/people/nikher

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
