## [Review Process File · EMBO Molecular Medicine]

Ribonucleotide reductase inhibitors suppress SAMHD1 ara-CTPase activity enhancing cytarabine efficacy

Sean G Rudd, Nikolaos Tsesmetzis, Kumar Sanjiv, Cynthia BJ Paulin, Lakshmi Sandhow, Juliane Kutzner, Ida Hed Myrberg, Sarah S Buntun, Hanna Axelsson, Si Min Zhang, Azita Rasti, Petri Mäkelä, Si'Ana A Coggins, Sijia Tao, Sharda Suman, Rui Mamede Branca, Georgios Mermelekas, Elisée Wiita, Sun Lee, Julian Waldfredsson, Raymond F Schinazi, Baek Kim, Janne Lehtiö, Georgios Rassidakis, Katja Pokrovskaja Tamm, Ulrika Warpman-Berglund, Mats Heyman, Dan Grandér, Sören Lehmann, Thomas Lundbäck, Hong Qian, Jan-Inge Henter, Torsten Schaller, Thomas Helleday, & Nikolas Herold

Review timeline:

Submission date:	1 February 2019
Editorial Decision:	12 March 2019
Authors appealed:	13 March 2019
Editorial Decision:	21 March 2019
Revision received:	5 November 2019
Editorial Decision:	28 November 2019
Revision received:	11 December 2019
Editorial Decision:	13 December 2019
Revision received:	15 December 2019
Accepted:	17 December 2019

Editor: Lise Roth

Transaction Report:

1st Editorial Decision

12 March 2019

Thank you for submitting your manuscript "Ribonucleotide reductase inhibitors suppress SAMHD1 ara-CTPase activity enhancing ara-C efficacy" to EMBO Molecular Medicine. We have now received the comments from the two reviewers who agreed to evaluate your manuscript.

As you will see from the enclosed reports, while they both acknowledge interesting findings in your study, they also have serious and partly overlapping issues. Considering the substantial points raised and the overall rather low level of support provided by the reviewers, I am afraid I see little choice but to return the manuscript to you at this point with the decision that we cannot offer to publish it.

***** Reviewer's comments *****

Referee #1 (Remarks for Author):

In this paper Rudd et al explore the potential synergism between cytarabine and other antimetabolites (such as gemcitabine, hydroxyurea and triapine) for the treatment of AML. They used a combination of in vitro screening tests, dose response matrices, synergism experiments, in vivo mouse models, and patient samples to demonstrate that molecules that act as RNR inhibitors improve Ara-C efficacy by indirect inhibition of SAMHD1, an enzyme that inactivates Ara-C active metabolites.

While some of the findings reported in the manuscript are interesting, the following major concerns exist:

1. The authors report a screening of 33000 compounds to look for hit that could inactivate SAMHD1, but they state the details of the screen will be published separately. Since the results have not yet been published, the authors should at least provide details on how the screen was performed, and how the top hit was chosen. The inability to evaluate the quality of the screen makes it difficult to evaluate the validity of the selected compounds.
2. Figure 1 C: the specific concentration of each of the RNR inhibitors tested should be reported in the figure legend next to each graph.
3. Figure 1C: it is concerning to see that the doses of RNRi that have the most effect in reducing EC50 of Ara-C, are toxic also in the controls (cells exposed to the RNRi but not treated with Ara-C). This is another reason that highlights the importance of adding the range of concentrations used for the RNRi. Alternatively, the authors could simplify this figure by selecting the doses that are simultaneously non-toxic to the controls in the absence of Ara-C treatment, and which also have the most synergistic effect.
4. Figure 2 is surprising, since THP1 cells have traditionally been unable to induce a lethal malignancy in immunocompromised mice. The authors should authenticate their cell lines, if they have not already done so.
5. The authors state that small molecules inhibitors of SAMHD1 have been reported previously, but have not demonstrated cell activity. How do they explain that an indirect inhibition of the same SAMHD1 through RNR inhibition is now an effective strategy? Would not be more likely that the two mechanisms are independent of one other?
6. It is unclear why the authors initially test for synergism with gemcitabine (based on the drug screen of SAMHD1 inhibitors) but then switch their focus to hydroxyurea based solely on speculating that this is an indirect inhibitor of SAMHD1.
7. Ara-C is an effective treatment for inducing remission in AML in combination with an anthracycline. If the authors argue that addition of HU would improve Ara-C therapy, they should demonstrate that the addition of HU can provide further benefit when added to an Ara-C and anthracycline combination.
8. The Methods reveal that the authors exposed cells to the RNRi for 72 hrs at different concentrations, and subsequently add Ara-C. This is a sequential treatment, rather than simple combination therapy. Importantly, the sequential use of hydroxyurea and cytarabine is not a new idea, nor is the concept of synergism between the two drugs (with the same mechanism proposed here); these findings were reported more than 30 years ago (https://doi.org/10.20772/cancersci1985.76.8_729, and <https://www.ncbi.nlm.nih.gov/pubmed/3819808>), but are not cited in this report.

Referee #2 (Comments on Novelty/Model System for Author):

The investigators seek to improve the efficacy of cytarabine, one of the most useful anti-leukemic agents, but identifying SAMHD1 and its effects on dNTP pools.

1. There should be some greater description of the screening asset for sensitizing SAMHD1 (authors states that it will be published elsewhere).
2. Fig 1b there are two separate horizontal panels for SAMHD1 -- what is the meaning of it. Legend is skimpy. RRM1,2,2b should be spelled out.
3. More than 1 AML cell line should be studied in Figure 1c and thereafter. There may be something unique about THP1 cell line as suggested by 1e.
4. Figures are not well discussed in text and there is a lot of back and forth of figure 1 components.
5. Fig EV4a, b should be part of the text.
6. Clinical data are from public databases (TCGA and TARGET). Primary tissue would be more compelling.
7. Fig 4 legend states "g,h", should be "alb"

Authors appealed

13 March 2019

Thank you for the decision letter on our manuscript EMM-2019-10419. You have decided to reject the manuscript based on "serious and partly overlapping issues" and "the substantial points raised

and the overall rather low level of support provided by the reviewers". However, based on the comments provided by reviewer 1 and 2, we see no scientific justification for that decision. Furthermore, the editorial guidelines of EMBO Molecular Medicine preclude additional referee comments that might explain your decision: "To further facilitate transparency, the journal has removed the "Confidential Comments" field from our referee reporting forms. This is to ensure that the authors receive all information pertinent to the decision made on a manuscript." Thus, to reiterate, we see no scientific basis for the editorial decision.

To give an overview of the reviewers points' (a detailed point-by-point can be found further down) - four out of eight points (4, 5, 6 and 8) raised by reviewer 1 stem from severe misreading and misinterpretation of data and information provided in our manuscript, as well as an apparent lack of meticulousness. At maximum, two out of eight points are mere copy-edit changes (2, 3) while only the remaining two out of eight points require additional data and/or experiments (1, and 7) that can be provided in due time and do not warrant the qualifier "major". Reviewer 2 raises two (out of seven points) that stem from misreading or misinterpretation of the manuscript (3, and 6), four out of seven points that are purely copy-edit in nature (2, 4, 5, and 7), and only one of seven points that requires additional data and/or experiments (1) that can be provided in due time and also does not warrant the qualifier "major". We would furthermore like to point out that only one issue raised by the reviewers was overlapping.

Based on the editorial guidelines of EMBO Molecular Medicine that "Authors may appeal decisions if there is concrete evidence for a misunderstanding or mistake at the editorial or referee level. Appeals are evaluated in depth and without prejudice.", we would therefore like to appeal your decision and suggest that you give us the opportunity to revise our manuscript with the minor revisions that would result from the reviewers' comments.

2nd Editorial Decision

21 March 2019

Thank you for your e-mail asking us to reconsider our decision on your manuscript. As mentioned in my previous email, I have asked a third reviewer to evaluate your manuscript.

As you will see from the report below, this referee acknowledges the potential medical impact of your study, but nevertheless has fundamental concerns that should be addressed in a major round of revision of the present manuscript, so that the data fully support the conclusions. Addressing the reviewers' concerns in full (including from referees #1 and #2) will be necessary for further considering the manuscript in our journal. EMBO Molecular Medicine encourages a single round of revision only and therefore, acceptance or rejection of the manuscript will depend on the completeness of your responses included in the next, final version of the manuscript.

***** Reviewer's comments *****

Referee #3 (Comments on Novelty/Model System for Author):

systems are adequate

Referee #3 (Remarks for Author):

In the present study the authors show that the dNTPase SAMHD1, which regulates dNTP homeostasis antagonistically to ribonucleotide reductase (RNR), limits ara-C efficacy by hydrolysing the active triphosphate metabolite ara-CTP. The authors identify that, that clinically used inhibitors of RNR, such as gemcitabine and hydroxyurea, overcome the SAMHD1-mediated barrier to ara-C efficacy in primary blasts and xenograft models of AML, displaying SAMHD1-dependent synergy with ara-C. They suggest that this is mediated by dNTP pool imbalances leading to allosteric reduction of SAMHD1 ara-CTPase activity. They suggest that SAMHD1 may stand as a novel biomarker for combination therapies of ara-C and RNR inhibitors with immediate consequences for clinical practice to improve treatment of AML.

These data might indeed become of value both for AML patient stratification as well as for novel therapeutic attempts.

Nevertheless, different points should be addressed.

1. Toxicity studies of the new combination should be included in the studies.
2. Data for SAMHD1 biomarker should be implemented not only using the TCGA AML data set but also the blueprint data set.
3. Furthermore, it seems that SAMHD1 expression might represent a marker modulating ARA-C resistance. Data using data set of ARA-C resistance might corroborate the correlation. Furthermore, if available, primary AML blasts sensitive and resistant to ARA-C should be included.
4. The in vivo data should be implemented by using mouse leukemia models or at least immune-competent mice.
5. Mechanistically speaking, it stays unclear whether SAMHD1 expression regulation in ARA-C resistant AML might derive as a red-out of the use of ARA-C. If so, which mechanism might be hypothesised? Is there a deregulation of the level of chromatin?
6. Experiments using hydroxyurea should also be implemented by using gemcitabine (dF-dC) which is a drug used frequently against cancer and might likely have an immediate redout.
7. Finally, the authors have successfully demonstrated that, in cell, combination of ara-C and dF-dC, HU and 3-AP have a synergistic effect. Then, for in vivo experiment they select the ara-C/HU combination. This choice is dictated by the fact that HU is currently employed to treat AML, is less toxic than dF-dC and also cheaper. On the other hand, it should be mentioned that HU has the potential to also inhibit other metal-containing enzymes including such as HDAC, carboxypeptidase A, urease, carbonic anhydrase and redox enzymes such as lipo-oxygenase thanks to its hydroxamate portion. Therefore, the efficacy of the inspected combination could be ascribed to the inhibition of RNR but also to a multiplicity of known targets. Thus it is suggested that the authors, perform experiments to rule out that HDAC inhibition by HU is a concurrent factor for the observed anticancer efficacy of the ara-C/HU. In this respect, the ara-C/dF-dC combination could be also explored in vivo.

2nd Revision - authors' response

5 November 2019

Point-by-point response for manuscript EMM-2019-10419-V3

We would first like to thank referees #3 and #4 for their comments. We also hope that the revisions based upon the comments of referees #1 and #2, for which a considerable amount of time and resources were spent, have been perceived satisfactory.

We believe that the additional comments by referee #4 led to important considerations regarding the mechanism by which RNRi cause an apparent loss of SAMHD1 enzymatic activity towards ara-CTP. Detailed point-by-point response can be found below.

Referee #3:

Remarks for Author

The present revised version of the article is very much improved. The authors have included data in support of their hypothesis.

We would like to thank the referee for this assessment. The improvement of the revised manuscript was only possible due to the valuable critique of the initially submitted manuscript.

Referee #4:

Comments on Novelty/Model System for Author

Novelty concern: The novelty of this work is the finding that combination therapy using ara-C with RNR inhibitors such as 3-AP, gemcitabine, and/or hydroxyurea that result in feedback inhibition of SAMHD1 lead to synergistic cytotoxicity in AML cells. However, SAMHD1 has been previously demonstrated by multiple groups to be the major enzyme responsible for inactivating the phosphorylated, active form of ara-C. Novelty could be substantially improved by identifying

precisely how RNR inhibitors indirectly block SAMHD1 activity (it appears to involve occupancy of the AS2 site of SAMHD1).

We agree with the referee that SAMHD1 is known to be a major factor leading to reduced efficacy of ara-C therapies by detoxifying cancer cells from ara-CTP, which we and others reported previously (Herold et al, 2017a; Herold et al, 2017b; Herold et al, 2017c; Hollenbaugh et al, 2017; Rassidakis et al, 2018; Schneider et al, 2017). However, identification of SAMHD1 as a factor that limits clinical efficacy of ara-C treatments is not sufficient to improve patient outcome. The overall aim of our study was to overcome SAMHD1-mediated ara-C resistance *in vivo* using a small molecule. No such molecules have been reported to date. Thus, the novelty of our study is the identification of clinically used cancer drugs (inhibitors of ribonucleotide reductase, RNRi) as a viable strategy to alleviate this SAMHD1-mediated resistance to ara-C therapy. We believe that these findings are not only novel, but exhibit a high translational impact as they could be directly implemented to treat patients given that at least one RNR inhibitor (RNRi), hydroxyurea, already has an indication for haematological malignancies.

We appreciate the concern that the mechanism of how RNRi indirectly lead to loss of SAMHD1 ara-CTPase activity has not been fully elucidated in our manuscript, and we explicitly state that future work is required. Nevertheless, we would like to stress that our manuscript contains a substantial amount of work in developing our initial identification of this phenomenon in an unbiased phenotypic screen through to providing a detailed model. Firstly, we identify gemcitabine (dF-dC) in a screen as a SAMHD1-dependent ara-C sensitizer and subsequently identify the target of dF-dC responsible for this phenotype to be RNR, given we can recapitulate this phenomenon with chemically distinct RNRi, HU and 3-AP. Secondly, we show that none of the RNRi or their metabolites directly inhibit SAMHD1 *in vitro* (Figure EV1D), and furthermore, using a cellular biophysical assay (CETSA), that RNRi do not bind to SAMHD1 in living cells (Figure EV1E and Figure EV5E-G). Thirdly, we show that RNRi do not lead to impaired tetramerisation of SAMHD1 in cells (Figure EV5A-D), which could be one explanation for loss of activity. Fourthly, we show that a specific class of RNRi, the allosteric purine nucleosides clofarabine, fludarabine and cladribine do not – unlike their non-allosteric RNRi counterparts HU, dF-dC and 3-AP – cause phenotypic loss of SAMHD1 ara-CTPase activity (Figure EV3). Those negative findings substantially narrow down the mechanism, in addition to providing data of relevance to AML treatment given some of these purine nucleosides are clinically combined with ara-C.

Next, we showed that apparent loss of SAMHD1 ara-CTPase activity due to treatment with RNRi coincided with a significant change in the relative composition of dNTP pools: with the main allosteric activator at allosteric site 2 (AS2) under physiological conditions, dATP, being reduced, whilst dCTP was increased, leading to an inverted ratio of these nucleotides in cells (Figure 4A, B and Figure S11A, B). We further show that this finding can, at least partially, be explained by activation of the dNTP salvage pathway through phosphorylation of dCK at serine-74 (Figure 4C), all of which we show with three chemically distinct RNRi. Eventually, we show in a biochemical assay that GTP:dCTP-activated SAMHD1 loses its ability to hydrolyse ara-CTP (Figure 4D). These findings allowed the generation of a model in which RNRi treatment leads to a shift of GTP:dATP SAMHD1 to GTP:dCTP SAMHD1, resulting in loss of ara-CTPase activity in cells (Figure EV5H).

In a newly revised manuscript, we can provide additional lines of evidence for the mechanism of RNRi-mediated loss of SAMHD1 ara-CTPase activity:

- 1) SAMHD1 can be post-translationally modified by reversible oxidation at cysteine residues. Oxidation is mediated by reactive oxygen species (ROS), resulting in inhibition of tetramerisation and enzymatic activity of SAMHD1 (Mauney et al, 2017). Since RNR inhibition by HU or dF-dC induces substantial ROS production (Patra et al, 2019; Somyajit et al, 2017), we hypothesised that RNRi-mediated SAMHD1-dependent sensitisation to ara-C might be mediated by inactivation of SAMHD1 through oxidation. To this end, we compared the effect of the ROS scavenger *N*-acetylcysteine (NAC) on the ability of HU to sensitise SAMHD1-proficient THP-1 cells to ara-C. However, NAC had no effect on synergy of HU and ara-C. These findings argue against a role of RNRi-induced ROS for SAMHD1-dependent synergy of RNRi and ara-C.

- 2) Another post-translational modification of SAMHD1 is threonine phosphorylation at position 592. T592 phosphorylation has been suggested to reduce the ability of SAMHD1 to hydrolyse dNTPs when dNTP levels are low (Arnold et al, 2015). In addition, T592 phosphorylation has been reported to regulate SAMHD1's substrate specificity (Jang et al, 2016), which could be of relevance to the phenomenon we report here. HU can perturb cell cycle progression, and thereby affect expression and/or activity of Cdk1 and Cdk2 (Rieber & Rieber, 1994; Rodriguez-Bravo et al, 2007; Tanguay & Chiles, 1994), the serine/threonine kinases responsible for SAMHD1 T592 phosphorylation (Yan et al, 2015). Hence, SAMHD1 phosphorylation might be mechanistically linked in RNRi-mediated reduction of SAMHD1 enzymatic activity towards ara-CTP. To investigate this, we determined the extent of HU-ara-C or dF-dC-ara-C synergy in THP-1 cells expressing either wildtype SAMHD1, a phosphomimetic T592E SAMHD1, or a non-phosphorylatable T592A SAMHD1. However, the extent of sensitisation to ara-C by either HU or dF-dC was largely unaffected by T592 mutations.

- 3) In addition to CETSA experiments that showed that thermostability of SAMHD1 is not affected by treatment with HU or dF-dC (Figure EV5E, F), indicating no substantial difference in the oligomeric state of SAMHD1, we performed *in vitro* thermal shift assays (differential scanning fluorimetry, DSF) of recombinant SAMHD1 in the presence of allosteric activators: either GTP:dATP α S or GTP:dCTP α S. The thermostability profile in the presence of dATP α S was indistinguishable from the one in the presence of dCTP α S. Taken together with our *in vitro* SAMHD1 enzyme activity assay that showed loss of ara-CTPase activity when using dCTP α S as an allosteric activator (Figure 4D), this suggests

that GTP:dCTP α S induces tetramerisation of SAMHD1, but that those tetramers have lost ara-CTPase activity (in contrast to GTP:dATP α S-induced tetramers). This is consistent with our CETSA experiments showing identical melting temperatures of SAMHD1 in RNRI-treated as compared to untreated cells (Figure EV 5E, F), even though RNRI treatment phenotypically abolishes ara-CTPase activity (Figure 1C-E).

Activation of SAMHD1 with GTP:dAPT α S or GTP:dCTP α S leads to similar changes in SAMHD1 structure in differential scanning fluorimetry (DSF) assays. (A) dATP α S or dCTP α S similarly altered the SAMHD1 melting curves, alone or combined with GTP. Recombinant human SAMHD1 protein (5 μ M) was treated with various concentrations of dATP α S or dCTP α S, alone or combined with GTP, for 30min before heat-induced denaturation been recorded by DSF. Mean relative fluorescence units (RFU) of replicate wells (3-4) shown. **(B)** Melting temperatures of recombinant SAMHD1 proteins as treated in A are summarized.

- 4) Next, using our *in vitro* activity assay with recombinant SAMHD1, we confirmed that GTP:dCTP SAMHD1 still harbours catalytic activity towards dCTP, as reported previously (Koharudin et al, 2014). However, at concentrations for which GTP:dCTP SAMHD1 hydrolyses dCTP (serving both as allosteric activator and substrate), no ara-CTP activity is observed in the GTP:dCTP α S SAMHD1. These findings further corroborate that these concentrations of dCTP (and so presumably dCTP α S) are indeed capable of inducing enzymatically active tetramers, which, however, have lost ara-CTPase activity. That SAMHD1 indeed might have differential substrate specificity is illustrated by the fact that high SAMHD1 expression in macrophages leads to consistent reduction of dATP, dGTP, dCTP, and dTTP while dUTP levels remain high – even though dUTP is a strong allosteric activator of SAMHD1 itself (Hansen et al, 2014; Kennedy et al, 2011).

GTP:dCTPaS bound SAMHD1 lacks ara-CTPase activity

Recombinant SAMHD1 was incubated with the indicated nucleotide combinations (each 200 μ M) in the enzyme-coupled malachite green assay. Absorbance at 630 nm indicates liberated inorganic phosphate, the product of nucleotide hydrolysis. Representative of 2 experiments shown, each performed in triplicate.

- 5) It has been reported that different allosteric activators – while retaining the overall structural properties of tetrameric SAMHD1 – can induce subtle conformational changes in SAMHD1. E.g., the histidine-215 side chain in the catalytic site of SAMHD1 is positioned differently in GTP:dATP SAMHD1 as compared to dGTP:dATP SAMHD1 (Koharudin et al, 2014). Future studies, though beyond the scope of this report, with the aim to resolve the co-crystals of GTP:dCTP:ara-CTP SAMHD1 and GTP:dATP:ara-CTP are required to answer the question whether conformational changes (e.g. in critical histidines of the allosteric site) lead to steric hindrance for ara-CTP hydrolysis. This might be a challenging endeavour though given that dCTP and dATP will compete with ara-CTP at the catalytic site.

Adequacy of model system concern: Choice of THP-1 cells as the primary cancer cell line for the initial screen appears to be made rather arbitrarily. It would be better to choose an AML cell line representative of molecular subtype(s) of responsive and non-responsive patients. This concern is mostly allayed by use of other leukemia cell lines in follow-up studies.

We agree that a screen with multiple cell lines with different genetic properties would be interesting in order to compare putative hit lists from a phenotypic screen. However, as we pointed out before, the actual screen is only auxiliary for the present study. As the referee acknowledges, we tried to validate our screening hit by testing multiple different cell lines and primary patient cells.

Remarks for Author

Substantial amounts of work have clearly been done over the course of the submission/revision process. Novelty of the findings are relatively low due to the obvious/straightforward finding that inhibiting SAMHD1 increases ara-C-mediated cytotoxicity in AML cells. Nevertheless, the finding that a certain class of RNR inhibitor can ultimately lead to inhibition of SAMHD1 and potentiate ara-C cytotoxicity in AML is of great clinical significance. This has the potential to inform clinical practice in the population of AML patients refractory to (or relapsing after) front line combination chemotherapy. To fully realize this potential, however, it will be important to define the mechanism by which RNR inhibitors like 3-AP, hydroxyurea, and gemcitabine inhibit SAMHD1 (presumably through the AS2 site of the enzyme).

As discussed in more detail above, we disagree that the identification and characterisation of the pharmacological strategy to inactivate SAMHD1 ara-CTPase lacks novelty. We appreciate that the referee acknowledges the important clinical significance of our study. While the referee is correct that further studies would be informative to precisely understand the molecular mechanism, we believe that the translational impact of our findings is apparent in spite of that. Nevertheless, we now provide further data ruling out the potential involvement of distinct post-translational modifications of SAMHD1, provide more evidence that GTP:dCTP activated SAMHD1 forms tetramers with enzymatic activity, albeit with a striking loss of ara-CTPase activity, and suggest structural biology studies that are beyond the scope of the current manuscript (see above).

References

- Arnold LH, Groom HC, Kunzelmann S, Schwefel D, Caswell SJ, Ordonez P, Mann MC, Rueschenbaum S, Goldstone DC, Pennell S et al (2015) Phospho-dependent Regulation of SAMHD1 Oligomerisation Couples Catalysis and Restriction. *PLoS Pathog* 11: e1005194
- Hansen EC, Seamon KJ, Cravens SL, Stivers JT (2014) GTP activator and dNTP substrates of HIV-1 restriction factor SAMHD1 generate a long-lived activated state. *Proc Natl Acad Sci U S A* 111: E1843-1851
- Herold N, Rudd S, Sanjiv K, Kutzner J, Bladh J, Paulin CBJ, Helleday T, Henter J-I, Schaller T (2017a) SAMHD1 protects cancer cells from various nucleoside-based antimetabolites. *Cell Cycle*
- Herold N, Rudd SG, Ljungblad L, Sanjiv K, Myrberg IH, Paulin CB, Heshmati Y, Hagenkort A, Kutzner J, Page BD et al (2017b) Targeting SAMHD1 with the Vpx protein to improve cytarabine therapy for hematological malignancies. *Nat Med* 23: 256-263
- Herold N, Rudd SG, Sanjiv K, Kutzner J, Myrberg IH, Paulin CBJ, Olsen TK, Helleday T, Henter JI, Schaller T (2017c) With me or against me: Tumor suppressor and drug resistance activities of SAMHD1. *Experimental hematology*
- Hollenbaugh JA, Shelton J, Tao S, Amiralaei S, Liu P, Lu X, Goetze RW, Zhou L, Nettles JH, Schinazi RF et al (2017) Substrates and Inhibitors of SAMHD1. *PLoS One* 12: e0169052
- Jang S, Zhou X, Ahn J (2016) Substrate specificity of SAMHD1 triphosphohydrolase activity is controlled by deoxyribonucleoside triphosphates and phosphorylation at Thr592. *Biochemistry*
- Kennedy EM, Daddacha W, Slater R, Gavegnano C, Fromentin E, Schinazi RF, Kim B (2011) Abundant non-canonical dUTP found in primary human macrophages drives its frequent incorporation by HIV-1 reverse transcriptase. *J Biol Chem* 286: 25047-25055
- Koharudin LM, Wu Y, DeLucia M, Mehrens J, Gronenborn AM, Ahn J (2014) Structural basis of allosteric activation of sterile alpha motif and histidine-aspartate domain-containing protein 1 (SAMHD1) by nucleoside triphosphates. *J Biol Chem* 289: 32617-32627
- Mauney C, Rogers L, Harris R, Daniel L, Devarie-Baez N, Wu H, Furdui C, Poole LB, Perrino F, Hollis T (2017) The SAMHD1 dNTP triphosphohydrolase is controlled by a redox switch. *Antioxidants & Redox Signaling*
- Patra KK, Bhattacharya A, Bhattacharya S (2019) Molecular dynamics investigation of a redox switch in the anti-HIV protein SAMHD1. *Proteins* 87: 748-759
- Rassidakis GZ, Herold N, Myrberg IH, Tsesmetzis N, Rudd SG, Henter JI, Schaller T, Ng SB, Chng WJ, Yan B et al (2018) Low-level expression of SAMHD1 in acute myeloid leukemia (AML) blasts correlates with improved outcome upon consolidation chemotherapy with high-dose cytarabine-based regimens. *Blood Cancer J* 8: 98
- Rieber M, Rieber MS (1994) Cyclin-dependent kinase 2 and cyclin A interaction with E2F are targets for tyrosine induction of B16 melanoma terminal differentiation. *Cell growth & differentiation : the molecular biology journal of the American Association for Cancer Research* 5: 1339-1346
- Rodriguez-Bravo V, Guaita-Esteruelas S, Salvador N, Bachs O, Agell N (2007) Different S/M checkpoint responses of tumor and non tumor cell lines to DNA replication inhibition. *Cancer research* 67: 11648-11656
- Schneider C, Oellerich T, Baldauf HM, Schwarz SM, Thomas D, Flick R, Bohnenberger H, Kaderali L, Stegmann L, Cremer A et al (2017) SAMHD1 is a biomarker for cytarabine response and a therapeutic target in acute myeloid leukemia. *Nat Med* 23: 250-255

Somyajit K, Gupta R, Sedlackova H, Neelsen KJ, Ochs F, Rask MB, Choudhary C, Lukas J (2017) Redox-sensitive alteration of replisome architecture safeguards genome integrity. *Science* 358: 797-802

Tanguay DA, Chiles TC (1994) Cell cycle-specific induction of Cdk2 expression in B lymphocytes following antigen receptor cross-linking. *Mol Immunol* 31: 643-649

Yan J, Hao C, DeLucia M, Swanson S, Florens L, Washburn MP, Ahn J, Skowronski J (2015) CyclinA2-Cyclin-dependent Kinase Regulates SAMHD1 Protein Phosphohydrolase Domain. *J Biol Chem* 290: 13279-13292

3rd Editorial Decision

28 November 2019

Thank you for submitting your revised manuscript to EMBO Molecular Medicine. As you remember, your manuscript had been originally sent to 2 reviewers, and based on their reports, the manuscript was rejected. You appealed this decision, and I consulted with a third reviewer, who recognized the high medical impact of your study. Major revisions were thus invited.

Unfortunately, none of the first 2 reviewers were available to re-review your manuscript, and I therefore contacted, together with reviewer #3, a 4th reviewer. This reviewer was alerted that this manuscript had already been reviewed and was asked to comment on the revisions.

I have now heard back from reviewer #3 and #4. As you will see from the reports below, reviewer #3 is now satisfied with the revisions and supports publication of the manuscript. Reviewer #4 raises 2 main points: the limited novelty and the lack of mechanistic understanding regarding SAMHD1 inhibition.

*****Reviewer's Comments*****

Referee #3:

Remarks for Author

The present revised version of the article is very much improved. The authors have included data in support of their hypothesis.

Referee #4:

Comments on Novelty/Model System for Author

Novelty concern: The novelty of this work is the finding that combination therapy using ara-C with RNR inhibitors such as 3-AP, gemcitabine, and/or hydroxyurea that result in feedback inhibition of SAMHD1 lead to synergistic cytotoxicity in AML cells. However, SAMHD1 has been previously demonstrated by multiple groups to be the major enzyme responsible for inactivating the phosphorylated, active form of ara-C. Novelty could be substantially improved by identifying precisely how RNR inhibitors indirectly block SAMHD1 activity (it appears to involve occupancy of the AS2 site of SAMHD1).

Adequacy of model system concern: Choice of THP-1 cells as the primary cancer cell line for the initial screen appears to be made rather arbitrarily. It would be better to choose an AML cell line representative of molecular subtype(s) of responsive and non-responsive patients. This concern is mostly allayed by use of other leukemia cell lines in follow-up studies.

Remarks for Author

Substantial amounts of work have clearly been done over the course of the submission/revision process. Novelty of the findings are relatively low due to the obvious/straightforward finding that inhibiting SAMHD1 increases ara-C-mediated cytotoxicity in AML cells. Nevertheless, the finding that a certain class of RNR inhibitor can ultimately lead to inhibition of SAMHD1 and potentiate ara-C cytotoxicity in AML is of great clinical significance. This has the potential to inform clinical practice in the population of AML patients refractory to (or relapsing after) front line combination chemotherapy. To fully realize this potential, however, it will be important to define the mechanism

by which RNR inhibitors like 3-AP, hydroxyurea, and gemcitabine inhibit SAMHD1 (presumably through the AS2 site of the enzyme).

3rd Revision - authors' response

11 December 2019

We would first like to thank referees #3 and #4 for their comments. We also hope that the revisions based upon the comments of referees #1 and #2, for which a considerable amount of time and resources were spent, have been perceived satisfactory.

We believe that the additional comments by referee #4 led to important considerations regarding the mechanism by which RNRi cause an apparent loss of SAMHD1 enzymatic activity towards ara-CTP. Detailed point-by-point response can be found below.

Referee #3:

Remarks for Author

The present revised version of the article is very much improved. The authors have included data in support of their hypothesis.

We would like to thank the referee for this assessment. The improvement of the revised manuscript was only possible due to the valuable critique of the initially submitted manuscript.

Referee #4:

Comments on Novelty/Model System for Author

Novelty concern: The novelty of this work is the finding that combination therapy using ara-C with RNR inhibitors such as 3-AP, gemcitabine, and/or hydroxyurea that result in feedback inhibition of SAMHD1 lead to synergistic cytotoxicity in AML cells. However, SAMHD1 has been previously demonstrated by multiple groups to be the major enzyme responsible for inactivating the phosphorylated, active form of ara-C. Novelty could be substantially improved by identifying precisely how RNR inhibitors indirectly block SAMHD1 activity (it appears to involve occupancy of the AS2 site of SAMHD1).

We agree with the referee that SAMHD1 is known to be a major factor leading to reduced efficacy of ara-C therapies by detoxifying cancer cells from ara-CTP, which we and others reported previously (Herold et al, 2017a; Herold et al, 2017b; Herold et al, 2017c; Hollenbaugh et al, 2017; Rassidakis et al, 2018; Schneider et al, 2017). However, identification of SAMHD1 as a factor that limits clinical efficacy of ara-C treatments is not sufficient to improve patient outcome. The overall aim of our study was to overcome SAMHD1-mediated ara-C resistance *in vivo* using a small molecule. No such molecules have been reported to date. Thus, the novelty of our study is the identification of clinically used cancer drugs (inhibitors of ribonucleotide reductase, RNRi) as a viable strategy to alleviate this SAMHD1-mediated resistance to ara-C therapy. We believe that these findings are not only novel, but exhibit a high translational impact as they could be directly implemented to treat patients given that at least one RNR inhibitor (RNRi), hydroxyurea, already has an indication for haematological malignancies.

We appreciate the concern that the mechanism of how RNRi indirectly lead to loss of SAMHD1 ara-CTPase activity has not been fully elucidated in our manuscript, and we explicitly state that future work is required. Nevertheless, we would like to stress that our manuscript contains a substantial amount of work in developing our initial identification of this phenomenon in an unbiased phenotypic screen through to providing a detailed model. Firstly, we identify gemcitabine (dF-dC) in a screen as a SAMHD1-dependent ara-C sensitizer and subsequently identify the target of dF-dC responsible for this phenotype to be RNR, given we can recapitulate this phenomenon with chemically distinct RNRi, HU and 3-AP. Secondly, we show that none of the RNRi or their metabolites directly inhibit SAMHD1 *in vitro* (Figure EV1D), and furthermore, using a cellular biophysical assay (CETSA), that RNRi do not bind to SAMHD1 in living cells (Figure EV1E and Figure EV5E-G). Thirdly, we show that RNRi do not lead to impaired tetramerisation of SAMHD1 in cells (Figure EV5A-D), which could be one explanation for loss of activity. Fourthly, we show that a specific class of RNRi, the allosteric purine nucleosides clofarabine, fludarabine and

cladribine do not – unlike their non-allosteric RNRi counterparts HU, dF-dC and 3-AP – cause phenotypic loss of SAMHD1 ara-CTPase activity (Figure EV3). Those negative findings substantially narrow down the mechanism, in addition to providing data of relevance to AML treatment given some of these purine nucleosides are clinically combined with ara-C.

Next, we showed that apparent loss of SAMHD1 ara-CTPase activity due to treatment with RNRi coincided with a significant change in the relative composition of dNTP pools: with the main allosteric activator at allosteric site 2 (AS2) under physiological conditions, dATP, being reduced, whilst dCTP was increased, leading to an inverted ratio of these nucleotides in cells (Figure 4A, B and Figure S11A, B). We further show that this finding can, at least partially, be explained by activation of the dNTP salvage pathway through phosphorylation of dCK at serine-74 (Figure 4C), all of which we show with three chemically distinct RNRi. Eventually, we show in a biochemical assay that GTP:dCTP-activated SAMHD1 loses its ability to hydrolyse ara-CTP (Figure 4D). These findings allowed the generation of a model in which RNRi treatment leads to a shift of GTP:dATP SAMHD1 to GTP:dCTP SAMHD1, resulting in loss of ara-CTPase activity in cells (Figure EV5H).

In the newly revised manuscript, we now provide additional lines of evidence for the mechanism of RNRi-mediated loss of SAMHD1 ara-CTPase activity:

1) SAMHD1 can be post-translationally modified by reversible oxidation at cysteine residues. Oxidation is mediated by reactive oxygen species (ROS), resulting in inhibition of tetramerisation and enzymatic activity of SAMHD1 (Mauney et al, 2017). Since RNR inhibition by HU or dF-dC induces substantial ROS production (Patra et al, 2019; Somyajit et al, 2017), we hypothesised that RNRi-mediated SAMHD1-dependent sensitisation to ara-C might be mediated by inactivation of SAMHD1 through oxidation. To this end, we compared the effect of the ROS scavenger N-acetylcysteine (NAC) on the ability of HU to sensitise SAMHD1-proficient THP-1 cells to ara-C. However, NAC had no effect on synergy of HU and ara-C. These findings argue against a role of RNRi-induced ROS for SAMHD1-dependent synergy of RNRi and ara-C.

A new section describing and discussing these findings (lines 320-325 and lines 421- 426) has been added to the main text together with a new Appendix Figure S11.

2) Another post-translational modification of SAMHD1 is threonine phosphorylation at position 592. T592 phosphorylation has been suggested to reduce the ability of SAMHD1 to hydrolyse dNTPs when dNTP levels are low (Arnold et al, 2015). In addition, T592 phosphorylation has been reported to regulate SAMHD1's substrate specificity (Jang et al, 2016), which could be of relevance to the phenomenon we report here. HU can perturb cell cycle progression, and thereby affect expression and/or activity of Cdk1 and Cdk2 (Rieber & Rieber, 1994; Rodriguez-Bravo et al, 2007; Tanguay & Chiles, 1994), the serine/threonine kinases responsible for SAMHD1 T592 phosphorylation (Yan et al, 2015). Hence, SAMHD1 phosphorylation might be mechanistically linked in RNRi-mediated reduction of SAMHD1 enzymatic activity towards ara-CTP. To investigate this, we determined the extent of HU-ara-C or dF-dC-ara-C synergy in THP-1 cells expressing either wildtype SAMHD1, a phosphomimetic T592E SAMHD1, or a non-phosphorylatable T592A SAMHD1. However, the extent of sensitisation to ara-C by either HU or dF-dC was largely unaffected by T592 mutations.

A new section describing and discussing these findings (lines 325-332 and lines 421- 426) has been added to the main text together with a new Appendix Figure S11.

3) Next, using our in vitro activity assay with recombinant SAMHD1, we confirmed that GTP:dCTP SAMHD1 still harbours catalytic activity towards dCTP, as reported previously (Koharudin et al, 2014). However, at concentrations for which GTP:dCTP SAMHD1 hydrolyses dCTP (serving both as allosteric activator and substrate), no ara-CTP activity is observed in the GTP:dCTPaS SAMHD1. These findings further corroborate that these concentrations of dCTP (and so presumably dCTPaS) are indeed capable of inducing enzymatically active tetramers, which, however, have lost ara-CTPase activity. That SAMHD1 indeed might have differential substrate specificity is illustrated by the fact that high SAMHD1 expression in macrophages leads to consistent reduction of dATP, dGTP, dCTP, and dTTP while dUTP levels remain high – even though dUTP is a strong allosteric activator of SAMHD1 itself (Hansen et al, 2014; Kennedy et al, 2011). New sections describing and discussing these findings (lines 387-390, lines 502-503, and 507-511) have been added to the main text together with a new Appendix Figure S13A.

4) In addition to CETSA experiments that showed that thermostability of SAMHD1 is not affected by treatment with HU or dF-dC (Figure EV5E, F), indicating no substantial difference in the oligomeric state of SAMHD1, we performed *in vitro* thermal shift assays (differential scanning fluorimetry, DSF) of recombinant SAMHD1 in the presence of allosteric activators: either GTP:dATPaS or GTP:dCTPaS. The thermostability profile in the presence of dATPaS was indistinguishable from the one in the presence of dCTPaS.

Taken together with our *in vitro* SAMHD1 enzyme activity assay that showed loss of ara-CTPase activity when using dCTPaS as an allosteric activator (Figure 4D), this suggests that GTP:dCTPaS induces tetramerisation of SAMHD1, but that those tetramers have lost ara-CTPase activity (in contrast to GTP:dATPaS-induced tetramers). This is consistent with our CETSA experiments showing identical melting temperatures of SAMHD1 in RNRi-treated as compared to untreated cells (Figure EV 5E, F), even though RNRi treatment phenotypically abolishes ara-CTPase activity (Figure 1C-E).

A new section describing these findings (lines 390-392) has been added to the main text together with a new Appendix Figure S13B and C.

5) It has been reported that different allosteric activators – while retaining the overall structural properties of tetrameric SAMHD1 – can induce subtle conformational changes in SAMHD1. E.g., the histidine-215 side chain in the catalytic site of SAMHD1 is positioned differently in GTP:dATP SAMHD1 as compared to dGTP:dATP SAMHD1 (Koharudin et al, 2014). Future studies, though beyond the scope of this report, with the aim to resolve the co-crystals of GTP:dCTP:ara-CTP SAMHD1 and GTP:dATP:ara-CTP are required to answer the question whether conformational changes (e.g. in critical histidines of the allosteric site) lead to steric hindrance for ara-CTP hydrolysis. This might be a challenging endeavour though given that dCTP and dATP will compete with ara-CTP at the catalytic site. A discussion point has been added accordingly in the main text (lines 511-516).

Adequacy of model system concern: Choice of THP-1 cells as the primary cancer cell line for the initial screen appears to be made rather arbitrarily. It would be better to choose an AML cell line representative of molecular subtype(s) of responsive and non-responsive patients. This concern is mostly allayed by use of other leukemia cell lines in follow-up studies.

We agree that a screen with multiple cell lines with different genetic properties would be interesting in order to compare putative hit lists from a phenotypic screen. However, as we pointed out before, the actual screen is only auxiliary for the present study. As the referee acknowledges, we tried to validate our screening hit by testing multiple different cell lines and primary patient cells.

Remarks for Author

Substantial amounts of work have clearly been done over the course of the submission/revision process. Novelty of the findings is relatively low due to the obvious/straightforward finding that inhibiting SAMHD1 increases ara-C-mediated cytotoxicity in AML cells. Nevertheless, the finding that a certain class of RNR inhibitor can ultimately lead to inhibition of SAMHD1 and potentiate ara-C cytotoxicity in AML is of great clinical significance. This has the potential to inform clinical practice in the population of AML patients refractory to (or relapsing after) front line combination chemotherapy. To fully realize this potential, however, it will be important to define the mechanism by which RNR inhibitors like 3-AP, hydroxyurea, and gemcitabine inhibit SAMHD1 (presumably through the AS2 site of the enzyme).

As discussed in more detail above, we disagree that the identification and characterisation of the pharmacological strategy to inactivate SAMHD1 ara-CTPase lacks novelty. We appreciate that the referee acknowledges the important clinical significance of our study. While the referee is correct that further studies would be informative to precisely understand the molecular mechanism, we believe that the translational impact of our findings is apparent in spite of that. Nevertheless, we now provide further data ruling out the potential involvement of distinct post-translational modifications of SAMHD1, provide more evidence that GTP:dCTP activated SAMHD1 forms tetramers with enzymatic activity, albeit with a striking loss of ara-CTPase activity, and suggest structural biology studies that are beyond the scope of the current manuscript (see above).

References

- Arnold LH, Groom HC, Kunzelmann S, Schwefel D, Caswell SJ, Ordonez P, Mann MC, Rueschenbaum S, Goldstone DC, Pennell S et al (2015) Phospho-dependent Regulation of SAMHD1 Oligomerisation Couples Catalysis and Restriction. *PLoS Pathog* 11: e1005194
- Hansen EC, Seamon KJ, Cravens SL, Stivers JT (2014) GTP activator and dNTP substrates of HIV-1 restriction factor SAMHD1 generate a long-lived activated state. *Proc Natl Acad Sci U S A* 111: E1843-1851
- Herold N, Rudd S, Sanjiv K, Kutzner J, Bladh J, Paulin CBJ, Helleday T, Henter J-I, Schaller T (2017a) SAMHD1 protects cancer cells from various nucleoside-based antimetabolites. *Cell Cycle*
- Herold N, Rudd SG, Ljungblad L, Sanjiv K, Myrberg IH, Paulin CB, Heshmati Y, Hagenkort A, Kutzner J, Page BD et al (2017b) Targeting SAMHD1 with the Vpx protein to improve cytarabine therapy for hematological malignancies. *Nat Med* 23: 256-263
- Herold N, Rudd SG, Sanjiv K, Kutzner J, Myrberg IH, Paulin CBJ, Olsen TK, Helleday T, Henter JI, Schaller T (2017c) With me or against me: Tumor suppressor and drug resistance activities of SAMHD1. *Experimental hematology*
- Hollenbaugh JA, Shelton J, Tao S, Amiralaei S, Liu P, Lu X, Goetze RW, Zhou L, Nettles JH, Schinazi RF et al (2017) Substrates and Inhibitors of SAMHD1. *PLoS One* 12: e0169052
- Jang S, Zhou X, Ahn J (2016) Substrate specificity of SAMHD1 triphosphohydrolase activity is controlled by deoxyribonucleoside triphosphates and phosphorylation at Thr592. *Biochemistry*
- Kennedy EM, Daddacha W, Slater R, Gavegnano C, Fromentin E, Schinazi RF, Kim B (2011) Abundant non-canonical dUTP found in primary human macrophages drives its frequent incorporation by HIV-1 reverse transcriptase. *J Biol Chem* 286: 25047-25055
- Koharudin LM, Wu Y, DeLucia M, Mehrens J, Gronenborn AM, Ahn J (2014) Structural basis of allosteric activation of sterile alpha motif and histidine-aspartate domain-containing protein 1 (SAMHD1) by nucleoside triphosphates. *J Biol Chem* 289: 32617-32627
- Mauney C, Rogers L, Harris R, Daniel L, Devarie-Baez N, Wu H, Furdai C, Poole LB, Perrino F, Hollis T (2017) The SAMHD1 dNTP triphosphohydrolase is controlled by a redox switch. *Antioxidants & Redox Signaling*
- Patra KK, Bhattacharya A, Bhattacharya S (2019) Molecular dynamics investigation of a redox switch in the anti-HIV protein SAMHD1. *Proteins* 87: 748-759
- Rassidakis GZ, Herold N, Myrberg IH, Tsesmetzis N, Rudd SG, Henter JI, Schaller T, Ng SB, Chng WJ, Yan B et al (2018) Low-level expression of SAMHD1 in acute myeloid leukemia (AML) blasts correlates with improved outcome upon consolidation chemotherapy with high-dose cytarabine-based regimens. *Blood Cancer J* 8: 98
- Rieber M, Rieber MS (1994) Cyclin-dependent kinase 2 and cyclin A interaction with E2F are targets for tyrosine induction of B16 melanoma terminal differentiation. *Cell growth & differentiation : the molecular biology journal of the American Association for Cancer Research* 5: 1339-1346
- Rodriguez-Bravo V, Guaita-Esteruelas S, Salvador N, Bachs O, Agell N (2007) Different S/M checkpoint responses of tumor and non tumor cell lines to DNA replication inhibition. *Cancer research* 67: 11648-11656
- Schneider C, Oellerich T, Baldauf HM, Schwarz SM, Thomas D, Flick R, Bohnenberger H, Kaderali L, Stegmann L, Cremer A et al (2017) SAMHD1 is a biomarker for cytarabine response and a therapeutic target in acute myeloid leukemia. *Nat Med* 23: 250-255

Somyajit K, Gupta R, Sedlackova H, Neelsen KJ, Ochs F, Rask MB, Choudhary C, Lukas J (2017) Redox-sensitive alteration of replisome architecture safeguards genome integrity. *Science* 358: 797-802

Tanguay DA, Chiles TC (1994) Cell cycle-specific induction of Cdk2 expression in B lymphocytes following antigen receptor cross-linking. *Mol Immunol* 31: 643-649

Yan J, Hao C, DeLucia M, Swanson S, Florens L, Washburn MP, Ahn J, Skowronski J (2015) CyclinA2-Cyclin-dependent Kinase Regulates SAMHD1 Protein Phosphohydrolase Domain. *J Biol Chem* 290: 13279-13292

4th Editorial Decision

13 December 2019

Thank you for the submission of your revised manuscript to EMBO Molecular Medicine. We have sent it to referee #4, and have now received his/her report. As you will see, this referee is now supportive of publication, and I am therefore very pleased to inform you that we will be able to accept your manuscript pending final editorial amendments.

***** Reviewer's comments *****

Referee #4 (Comments on Novelty/Model System for Author):

Novelty: It remains disappointing that, despite the authors' best efforts, they could not pinpoint the mechanism through which RNR inhibitors ultimately blocked SAMHD1 activity, which would have significantly boosted the novelty.

Referee #4 (Remarks for Author):

Although it is disappointing that the authors could not pinpoint the mechanism through which RNR inhibitors ultimately blocked SAMHD1 activity, they have done an outstanding job of addressing this concern by ruling out several known mechanisms influencing SAMHD1 activity. Furthermore, they have produced a logical and testable model based on supportive data of how SAMHD1 may be inhibited (altered enzyme conformation based on ratios of different deoxyribo- and ribonucleotides). This is all that can reasonably be asked of anyone and I have no issues with this manuscript's publication. Indeed, it should be published and the authors should be commended for their efforts (as well as their patience for dealing with a 4th reviewer).

4th Revision - authors' response

15 December 2019

Authors made the requested editorial changes.

Corresponding Author Name: Nikolas Herold

Manuscript Number: EMM-2019-10419-V3